# OPTIMAL SAMPLE COMPLEXITY OF CONTRASTIVE LEARNING

**Noga Alon**[1][*]**, Dmitrii Avdiukhin**[2]**, Dor Elboim**[3]**, Orr Fischer**[4][†]**, Grigory Yaroslavtsev**[5]

[1]Princeton University, [2]Northwestern University,
[3]Institute for Advanced Study, [4]Weizmann Institute of Science, [5]George Mason University,
`nalon@math.princeton.edu`, `dmitrii.avdiukhin@northwestern.edu`,
`dorelboim@gmail.com`, `orr.fischer@weizmann.ac.il`, `grigory@gmu.edu`

## ABSTRACT

Contrastive learning is a highly successful technique for learning representations of data from labeled tuples, specifying the distance relations within the tuple. We study the sample complexity of contrastive learning, i.e. the minimum number of labeled tuples sufficient for getting high generalization accuracy. We give tight bounds on the sample complexity in a variety of settings, focusing on arbitrary distance functions, both general $\ell_p$-distances, and tree metrics. Our main result is an (almost) optimal bound on the sample complexity of learning $\ell_p$-distances for integer $p$. For any $p \geq 1$ we show that $\tilde{\Theta}(\min(nd, n^2))$ labeled tuples are necessary and sufficient for learning $d$-dimensional representations of $n$-point datasets. Our results hold for an arbitrary distribution of the input samples and are based on giving the corresponding bounds on the Vapnik-Chervonenkis/Natarajan dimension of the associated problems. We further show that the theoretical bounds on sample complexity obtained via VC/Natarajan dimension can have strong predictive power for experimental results, in contrast with the folklore belief about a substantial gap between the statistical learning theory and the practice of deep learning.

## 1 INTRODUCTION

Contrastive learning (Gutmann & Hyvärinen, 2010) has recently emerged as a powerful technique for learning representations, see e.g. Smith & Eisner (2005); Mikolov et al. (2013); Dosovitskiy et al. (2014); Schroff et al. (2015a); Wang & Gupta (2015); Wu et al. (2018); Logeswaran & Lee (2018a); Hjelm et al. (2019); He et al. (2020); Tian et al. (2020); Chen et al. (2020); Chen & He (2021); Gao et al. (2021); Chen et al. (2021). In this paper we study the generalization properties of contrastive learning, focusing on its sample complexity. Despite recent interest in theoretical foundations of contrastive learning, most of the previous work approaches this problem from other angles, e.g. focusing on the design of specific loss functions (HaoChen et al., 2021), transfer learning (Saunshi et al., 2019; Chuang et al., 2020), multi-view redundancy (Tosh et al., 2021), inductive biases (Saunshi et al., 2022; HaoChen & Ma, 2023), the role of negative samples (Ash et al., 2022; Awasthi et al., 2022), mutual information (van den Oord et al., 2018; Hjelm et al., 2019; Bachman et al., 2019; Tschannen et al., 2020), etc. (Wang & Isola, 2020; Tsai et al., 2021; Zimmermann et al., 2021; von Kügelgen et al., 2021; Mitrovic et al., 2021; Wen & Li, 2021).

Generalization is one of the central questions in deep learning theory. However, despite substantial efforts, a folklore belief still persists that the gap between classic PAC-learning and the practice of generalization in deep learning is very wide. Here we focus on the following high-level question: *can PAC-learning bounds be non-vacuous in the context of deep learning*? Direct applications of PAC-learning to explain generalization in neural networks lead to vacuous bounds due to the high expressive power of modern architectures. Hence, recent attempts towards understanding

---

[*]Partially supported by NSF grant DMS-2154082

[†]Partially supported by the European Research Council (ERC) under the European Union's Horizon 2020 research and innovation programme (grant agreement No. 949083)

generalization in deep learning involve stability assumptions (Hardt et al., 2016), PAC-Bayesian bounds (Langford & Caruana, 2001; Dziugaite & Roy, 2017; Neyshabur et al., 2017), capacity bounds (Neyshabur et al., 2019), sharpness (Keskar et al., 2017; Lyu et al., 2022) and kernelization (Arora et al., 2019; Wei et al., 2019) among others (see e.g. Jiang et al. (2020) for an overview). In this paper, we make a step towards closing this gap by showing that the classic PAC-learning bounds have strong predictive power over experimental results.

We overcome this gap by changing the assumptions used to analyze generalization, shifting the emphasis from the inputs to the outputs. While it is typical in the literature to assume the presence of latent classes in the input (Saunshi et al., 2019; Ash et al., 2022; Awasthi et al., 2022), we avoid this assumption by allowing arbitrary input distributions. Instead, we shift the focus to understanding the sample complexity of training a deep learning pipeline whose output dimension is $d$, only assuming that by a suitable choice of architecture, one can find the best mapping of a given set of samples into $\mathbb{R}^d$. Hence, our assumptions about the architecture are very minimal, allowing us to study the sample complexity more directly. This is in contrast with the previous work, where a typical approach in the study of generalization (see e.g. Zhang et al. (2017)) is to assume that $n$ input points are in $\mathbb{R}^d$, while here we only assume the existence of an embedding of these points into a (metric) space such as $\mathbb{R}^d$. This avoids the challenges faced by Zhang et al. (2017) and related work, which shows that a two-layer network with $2n + d$ parameters can learn any function of the input.

## 1.1 OUR RESULTS

Consider a simple model for contrastive learning in which labeled samples $(x, y^+, z_1^-, \ldots, z_k^-)$ are drawn i.i.d. from an unknown joint distribution $\mathcal{D}$. [1] Here $x$ is referred to as the *anchor*, $y^+$ is a *positive* example and $z_i^-$ are *negative* examples, reflecting the fact that the positive example is "more similar" to the anchor than the negative examples. Before the tuple is labeled, it is presented as an anchor $x$ together with an unordered $(k+1)$-tuple $(x_1, \ldots, x_{k+1})$, from which the labeling process then selects a single positive example $y^+$ and $k$ negative examples $z_1^-, \ldots, z_k^-$.

The goal of contrastive learning is to learn a representation of similarity between the domain points, usually in the form of a metric space. For samples from a domain $V$, the learned representation is a distance function $\rho\colon V \times V \to \mathbb{R}$. A highly popular choice for the domain is the $\ell_p$-space under $\ell_p$-distance $\rho_p(x, y) = \|f(x) - f(y)\|_p$ for some $f\colon V \to \mathbb{R}^d$, with $p = 2$ used most frequently. Our central question is:

*How many contrastive samples are needed for learning a good distance function?*

The number of samples is a primary factor in the computational cost of training. While it is typical for the previous work to consider the number of samples required for generalization to be somewhat unimportant and focus on performance on the downstream tasks (in fact, contrastive learning is often referred to as an unsupervised learning method, see e.g. Saunshi et al. (2019)), here we focus specifically on the sample complexity. This is due to the fact that even if the samples themselves might be sometimes easy to obtain (e.g. when class labels are available and $(x, y^+)$ are sampled from the same class[2]), the computational cost is still of major importance and scales linearly with the number of samples.

Furthermore, in some settings, sample complexity might itself correspond to the cost of labeling, making the contrastive learning process directly supervised. Consider, for example, a crowdsourcing scenario (see e.g. Zou et al. (2015)), where a triple is shown as $(x, y, z)$ to a labeler who labels it according to whether $y$ or $z$ is more similar to $x$. In this setting, data collection is of primary importance since the sample complexity corresponds directly to the labeling cost.

In this paper, we state our main results first for the case $k = 1$ in order to simplify the presentation. Our bounds for general $k$ are obtained by straightforward modifications of the proofs for this case.

---

[1]This is substantially more general and hence includes as special cases typical models used in the literature which make various further assumptions about the structure of $\mathcal{D}$ (e.g. Saunshi et al. (2019); Chuang et al. (2020); Awasthi et al. (2022)).

[2]In this case, class information is still required, making this approach somewhat supervised. Supervision can be fully avoided, e.g. by creating a positive example from the anchor point (e.g. via transformations), although this can affect the quality of the learned metric due to the more limited nature of the resulting distribution.

Table 1: Sample complexity for contrastive learning for a dataset of size $n$. The notation $O_{\epsilon,\delta}$ hides dependence on $\epsilon$ and $\delta$. Dimension of the representation is denoted as $d$. Let $\tilde{d} = \min(n, d)$.

| Setting | Lower bound | Upper bound |
|---|---|---|
| $\ell_p$ for integer $p$ | $\Omega_{\epsilon,\delta}(n\tilde{d})$, Thm. 3.1 | Even $p$: $O_{\epsilon,\delta}(n\tilde{d})$ (matching), Thm. 3.2 |
| | | Odd $p$: $O_{\epsilon,\delta}(n\min(d\log n, n))$, Thm. 3.2 |
| | | Constant $d$: $O_{\epsilon,\delta}(n)$ (matching), Thm. 3.2 |
| $(1+\alpha)$-separate $\ell_2$ | $\Omega_{\epsilon,\delta}(n/\alpha)$, Thm. 4.2 | $\tilde{O}_{\epsilon,\delta}(n/\alpha^2)$, Thm. 4.2 |
| Arbitrary distance | $\Omega_{\epsilon,\delta}(n^2)$, Thm. 2.7 | $O_{\epsilon,\delta}(n^2)$ (matching), Thm. 2.7 |
| Cosine similarity | $\Omega_{\epsilon,\delta}(n\tilde{d})$ Thm D.3 | $O_{\epsilon,\delta}(n\tilde{d})$ (matching) Thm D.3 |
| Tree metric | $\Omega_{\epsilon,\delta}(n)$, Thm. D.2 | $O_{\epsilon,\delta}(n\log n)$, Thm. D.2 |
| **Generalizations, in the same settings** | | |
| Quadruplet learning | Same, Cor. C.3 | Same, Cor. C.3 |
| $k$ negatives | Same, Thm. 4.1,C.1 | Same, with extra $\log(k+1)$ factor, Thm. 4.1 |

**Theoretical Results** We address the above central question in the framework of PAC-learning (Valiant, 1984) by giving bounds on the number of samples required for the prediction of subsequent samples. Recall that we use notation $(x, y, z)$ to refer to unlabeled samples and $(x, y^+, z^-)$ for their labeled counterparts. In contrastive learning, a classifier is given access to $m$ labeled training samples $\{(x_i, y_i^+, z_i^-)\}_{i=1}^m$ drawn from $\mathcal{D}$. The goal of the classifier is to correctly classify new samples from $\mathcal{D}$. I.e. for every such sample with the ground truth label hidden and thus presented as $(x, y, z)$, to correctly label it as either $(x, y^+, z^-)$ or $(x, z^+, y^-)$. For the resulting classifier, we refer to the probability (over $\mathcal{D}$) of incorrectly labeling a new sample as the error rate.

Let $\mathcal{H}$ be a hypothesis class (in our case, metric spaces, $\ell_p$-distances, etc.). For parameters $\epsilon, \delta > 0$, the goal of the training algorithm is to find with probability at least $1 - \delta$ a classifier with the error rate at most $\epsilon + \epsilon^*$, where $\epsilon^*$ is the smallest error rate achieved by any classifier from $H$ (i.e. the best error achieved by a metric embedding, $\ell_p$-embedding, etc.). We refer to the number of samples required to achieve the required error rate as *sample complexity*. See Appendix A for a concrete example.

As is standard in PAC-learning, we state results in two settings: *realizable* and *agnostic*. Here realizable refers to the case when there exists a distance function in our chosen class (e.g. $\ell_2$-distance) that perfectly fits the data, i.e. $\epsilon^* = 0$, while in the agnostic setting, such a function might not exist. Let $n = |V|$ be the number of data points in the dataset, which the triples are sampled from. We summarize our results in Table 1. To simplify the presentation, for the rest of the section we assume that $\epsilon, \delta$ are fixed constants (see full statements of our results for the precise bounds, which show optimal dependence on these parameters). We first give a simple baseline, which characterizes the number of samples required for contrastive learning.

**Theorem 1.1** (Arbitrary distance[3] (informal), full statement: Theorem 2.7). *The sample complexity of contrastive learning for arbitrary distance functions is $\Theta(n^2)$, where the lower bound holds even for metric distances.*

This basic guarantee is already substantially better than the overall $\Theta(n^3)$ number of different triples, but can still be a pessimistic estimate of the number of samples required. We thus leverage the fact that in practice $\ell_p$-spaces are a typical choice for a representation of the distance function between the data points. In particular, $\ell_2$-distance (and the related cosine similarity) is most frequently used as a measure of similarity between points after embedding the input data into $\mathbb{R}^d$. Here the dimension $d$ of the embedding space is typically a fixed large constant depending on the choice of the architecture (e.g. $d = 512, 1024, 4096$ are popular choices).

Our main results are much stronger bounds on the sample complexity of learning representations under these $\ell_p$ distances. For any $\ell_p$-distance we get the following result:

---

[3]In this paper, the only requirement for arbitrary distances is symmetry, i.e. $\rho(x, y) = \rho(y, x)$

**Theorem 1.2** ($\ell_p$-distance upper bound (informal), full statement: Theorem 3.2). *For any constant integer p, the sample complexity of contrastive learning under $\ell_p$-distance is $O(\min(nd, n^2))$ if p is even, and $O(\min(nd \log n, n^2))$ if p is odd. Furthermore, if $d = O(1)$, then the sample complexity is $O(n)$ for any positive integer p.*

We show an $\Omega(\min(nd, n^2))$ lower bound for any $\ell_p$-distance, essentially matching our upper bounds:

**Theorem 1.3** ($\ell_p$-distance lower bound (informal), full statement: Theorem 3.1). *For any constant integer p, the sample complexity of contrastive learning under $\ell_p$-distance is $\Omega(\min(nd, n^2))$.*

For cosine similarity, we show the same result:

**Theorem 1.4** (Cosine similarity (informal), full statement: Theorem D.3). *The sample complexity of contrastive learning under cosine similarity is $\Theta(\min(nd, n^2))$.*

Using the techniques we develop for the above results, we also immediately get the following result:

**Theorem 1.5** (Tree distance (informal), full statement: Theorem D.2). *The sample complexity of contrastive learning under tree distance is $\Omega(n)$ and $O(n \log n)$.*

We also present multiple variations and extensions to complement the main results above. First, we show that the results above can be extended to the case where the positive example has to be selected among $k + 1$ options instead of just two, as is typical in the literature. This results in a $\log k$ increase in the sample complexity (see Theorem 4.1). Furthermore, our results can be extended to the case when the samples are well-separated, i.e. the distance to the negative example is at least a factor $(1 + \alpha)$ larger than the distance to the positive example. This is a frequent case in applications when the positive example is often sampled to be much closer to the anchor, which is often referred to as learning with hard negatives. In this case, we give an $\tilde{O}(n/\alpha^2)$ upper bound and an $\Omega(n/\alpha)$ lower bound (see Theorem 4.2).

Finally, we show in Corollary C.3 that all of our results can be adapted to another popular scenario in contrastive learning, where instead of sampling triples, one instead samples quadruples $((x_1^+, x_2^+), (x_3^-, x_4^-))$ with the semantic that the pair $(x_1^+, x_2^+)$ is more similar than the pair $(x_3^-, x_4^-)$. While the lower bounds transfer trivially, it is somewhat surprising that the upper bounds also hold for this case despite the fact that the overall number of different samples is $\Theta(n^4)$.

**Outline of the techniques** To describe the main idea behind our techniques, consider the case of $\ell_2$-metrics in a $d$-dimensional space. Let $f \colon V \to \mathbb{R}^d$ be a function mapping elements to their representations, and then the query $(x, y, z)$ is labeled $(x, y^+, z^-)$ if $\|f(x) - f(y)\|_2 < \|f(x) - f(z)\|_2$. This is equivalent to $\sum_{i=1}^{d}(f_i(x) - f_i(y))^2 < \sum_{i=1}^{d}(f_i(x) - f_i(z))^2$, where $f_i$ denotes the $i$-th coordinate of the representation. This condition can be written in the form $P(f_1(x), \ldots, f_d(x), f_1(y), \ldots, f_d(y), f_1(z), \ldots, f_d(z)) < 0$, where $P$ is some polynomial of degree 2. Hence, every input triplet corresponds to a polynomial such that the label of the triplet is decided by the sign of this polynomial. We show that the number of possible satisfiable sign combinations of the polynomials – and hence the largest shattered set of triplets – is bounded.

**Experimental results** To verify that our results indeed correctly predict the sample complexity, we perform experiments on several popular image datasets: CIFAR-10/100 and MNIST/Fashion-MNIST. We find the representations for these images using ResNet18 trained from scratch using various contrastive losses. Our experiments show that for a fixed number of samples, the error rate is well approximated by the value predicted by our theory. We present our findings in Appendix F.

## 1.2 OTHER PREVIOUS WORK

For a survey on metric learning we refer the reader the the monograph by Kulis (2013). A popular approach to metric learning involves learning Mahalanobis metrics, see Verma & Branson (2015) for PAC-like bounds in this setting. Wang & Tan (2018) study the sample complexity of this problem in the presence of label noise. We note that our bounds are more general since Mahalanobis distances only represent a subclass of possible metrics.

Related problems are also studied in the literature on ordinal embeddings, which requires adaptive queries in order to recover the underlying embedding. This is in contrast with our work where the samples are taken non-adaptive from an unknown distribution. For triplet queries, Arias-Castro (2017) shows that it's possible to recover the embedding from the triplets queries, up to a transformation, and provide convergence rates for quadruplet queries. Ghosh et al. (2019) shows that $O(d^8 n \log n)$ adaptive noisy triplet queries suffice for recovering an embedding up to a transformation with $O(1)$ additive divergence. Terada & Luxburg (2014) shows that it's possible to recover the ground-truth points up to a transformation assuming that these points are sampled from a continuous distribution assuming the kNN graph is given.

## 2 PRELIMINARIES

In order to state our main results, we first introduce formal definitions of contrastive learning in the realizable and agnostic setting and the associated notions of Vapnik-Chervonenkis and Natarajan dimension. We start by giving the definitions for the simplest triplet case.

**Definition 2.1** (Contrastive learning, realizable case). *Let $\mathcal{H}$ be a hypothesis class and $\rho \in \mathcal{H}$ be an unknown distance function.[4] Given access to samples of the form $(x, y^+, z^-)$ from a distribution $\mathcal{D}$, meaning $\rho(x, y) < \rho(x, z)$, the goal of contrastive learning is to create a classifier from $\mathcal{H}$ which accurately labels subsequent unlabeled inputs. For an error parameter $\epsilon \in (0, 1/2)$ [5], the sample complexity of contrastive learning, denoted as $S_3(\epsilon, \delta)$, is the minimum number of samples required to achieve error rate $\epsilon$ with probability at least $1 - \delta$.*

**Definition 2.2** (Contrastive learning, agnostic case). *Let $\mathcal{H}$ be a hypothesis class. Given access to labeled samples of the form $(x, y^+, z^-)$ from a distribution $\mathcal{D}$, the goal of agnostic contrastive learning is to create a classifier from $\mathcal{H}$ which accurately labels subsequent unlabeled inputs. For an error parameter $\epsilon > 0$, the sample complexity of contrastive learning, denoted as $S_3^a(\epsilon, \delta)$, is the minimum number of samples required to achieve error rate $\epsilon + \epsilon^*$ with probability at least $1 - \delta$, where $\epsilon^*$ is the best error rate that can be achieved by a hypothesis $\rho \in \mathcal{H}$.*

The above definitions can be naturally generalized to the case when instead of triplets we get inputs of the form $(x, x_1, \ldots, x_{k+1})$ and the objective is to select $x_i$ minimizing $\rho(x, x_i)$.

**Definition 2.3** (Quadruplet learning). *If instead of triplets as in the definitions above, the samples are quadruples of the form $((x, y), (z, w))$ labeled according to whether the $(x, y)$ pair is closer than the $(z, w)$ pair, we refer to this problem as quadruplet contrastive learning and denote the corresponding sample complexities as $S_Q(\epsilon, \delta)$ and $S_Q^a(\epsilon, \delta)$.*

**VC and Natarajan dimension** The main tool used for sample bounds is the Vapnik–Chervonenkis (VC) dimension for binary classification and the Natarajan dimension for multi-label classification.

**Definition 2.4** (VC-dimension (Vapnik & Chervonenkis, 1971)). *Let $X$ be the set of inputs. Let $H \subseteq 2^X$ be a set of subsets of $X$, called the hypothesis space. We say that $A \subset X$ is shattered by $H$ if for any $B \subset A$ there exists a hypothesis $h \in H$ such that $h \cap A = B$. Finally, the Vapnik–Chervonenkis (VC) dimension of $H$ is defined as the size of the largest shattered set.*

Intuitively, a set $A$ is shattered if all of $2^{|A|}$ possible labelings on $A$ are realizable by $H$. In our case, the set of inputs $X$ is defined by all possible triplets on $V$, i.e. $X \subseteq V^3$.

The set of distance functions $\Delta$ will vary depending on the setting. Below we show results when $\Delta$ contains 1) all possible distance functions[6], 2) all metrics induced by the $\ell_p$-norms over $n$ points in $\mathbb{R}^d$, and 3) other distance functions, such as tree metrics, cosine similarity etc.

For non-binary (finite) labels, the key property we use to characterize the sample complexity of a problem is the Natarajan dimension, which generalizes the VC dimension Ben David et al. (1995):

---

[4]The inputs are labeled based on $\rho$. With a slight abuse of notation, we refer to both the distance function and the hypothesis (i.e. a classifier) it defines as $\rho$.

[5]The $1/2$ error rate can be achieved by random guess, which requires zero samples. Hence, our theory makes prediction only for non-trivial values of $\epsilon$, as in Blumer et al. (1989)

[6]We only consider functions such that all distances are different. This is a reasonable assumption since in practice the probability of encountering exactly the same distances is zero. This assumption simplifies the analysis since we only need to consider the binary classification case

**Definition 2.5** (Natarajan dimension (Natarajan, 1989)). *Let $X$ be the set of inputs, $Y$ be the set of labels, and let $H \subseteq Y^X$ be a hypothesis class. Then $S \subseteq X$ is N-shattered by $H$ if there exist $f_1, f_2 \colon X \to Y$ such that $f_1(x) \neq f_2(x)$ for all $x \in A$ and for every $B \subseteq A$ there exists $g \in H$ such that:*

$$g(x) = f_1(x) \text{ for } x \in B \text{ and } g(x) = f_2(x) \text{ for } x \notin B$$

*Finally, the Natarajan dimension $\mathrm{Ndim}(H)$ of $H$ is the maximum size of an $N$-shattered set.*

**Lemma 2.6** ((Ben David et al., 1995), Informal). *If $|Y|$ is finite, then for the sample complexity $S(\epsilon, \delta)$ of the realizable case it holds that:*

$$S(\epsilon, \delta) = O\left(\frac{\mathrm{Ndim}(H)\log|Y|}{\epsilon}\operatorname{polylog}\left(\frac{1}{\epsilon}, \frac{1}{\delta}\right)\right) \text{ and } \Omega\left(\frac{\mathrm{Ndim}(H)}{\epsilon}\operatorname{polylog}\left(\frac{1}{\epsilon}, \frac{1}{\delta}\right)\right)$$

*In the agnostic case:*

$$S^a(\epsilon, \delta) = O\left(\frac{\mathrm{Ndim}(H)\log|Y|}{\epsilon^2}\operatorname{polylog}\left(\frac{1}{\epsilon}, \frac{1}{\delta}\right)\right) \text{ and } \Omega\left(\frac{\mathrm{Ndim}(H)}{\epsilon^2}\operatorname{polylog}\left(\frac{1}{\epsilon}, \frac{1}{\delta}\right)\right)$$

VC-dimension is a special case of Natarjan-dimension when $|Y| = 2$. In Appendix A we give examples to illustrate the notion of shattering.

Before presenting our main results, we state the following simple baseline, which we prove in Appendix D.

**Theorem 2.7** (Arbitrary distance). *For an arbitrary distance function $\rho \colon V \times V \to \mathbb{R}$ and a dataset of size $n$, the sample complexity of contrastive learning is $S_3(\epsilon, \delta) = \Theta\left(\frac{n^2}{\epsilon}\operatorname{polylog}\left(\frac{1}{\epsilon}, \frac{1}{\delta}\right)\right)$ in the realizable case and $S_3^a(\epsilon, \delta) = \Theta\left(\frac{n^2}{\epsilon^2}\operatorname{polylog}\left(\frac{1}{\epsilon}, \frac{1}{\delta}\right)\right)$ in the agnostic case. Furthermore, the lower bounds hold even if $\rho$ is assumed to be a metric.*

## 3  CONTRASTIVE LEARNING IN $\ell_p$-NORM

In this section, we show in most cases optimal bounds for the sample complexity of the contrastive learning problem for the $\ell_p$-metric in dimension $\mathbb{R}^d$, for any constant integer $p \geq 1$. First, we give the following lower bound, whose proof is deferred to Appendix B.1. Recall that $\rho_p(x, y) = \|f(x) - f(y)\|_p$ for $f \colon V \to \mathbb{R}^d$.

**Theorem 3.1** (Lower bound for $\ell_p$-distances). *For any real constant $p \in (0, +\infty)$, a dataset $V$ of size $n$, and the $\ell_p$ distance $\rho_p \colon V \times V \to \mathbb{R}$ in a $d$-dimensional space, the sample complexity of contrastive learning is $S_3(\epsilon, \delta) = \Omega\left(\frac{\min(nd, n^2)}{\epsilon}\operatorname{polylog}\left(\frac{1}{\epsilon}, \frac{1}{\delta}\right)\right)$ in the realizable case and $S_3^a(\epsilon, \delta) = \Omega\left(\frac{\min(nd, n^2)}{\epsilon^2}\operatorname{polylog}\left(\frac{1}{\epsilon}, \frac{1}{\delta}\right)\right)$ in the agnostic case.*

We then show that for integer $p$ this bound can be closely matched by the following upper bounds:

**Theorem 3.2** (Upper bound for $\ell_p$-distances for integer $p$). *For integer $p$, a dataset $V$ of size $n$, and the $\ell_p$ distance $\rho_p \colon V \times V \to \mathbb{R}$ in a $d$-dimensional space, the sample complexity of contrastive learning is upper-bounded as shown in Table 2.*

| Setting | Realizable case | Agnostic case |
|---|---|---|
| Even $p$ | $\min(nd, n^2)/\epsilon$ | $\min(nd, n^2)/\epsilon^2$ |
| Odd $p$ | $\min(nd\log n, n^2)/\epsilon$ | $\min(nd\log n, n^2)/\epsilon^2$ |
| Constant $d$ | $n/\epsilon$ | $n/\epsilon^2$ |

Table 2: Upper bounds on sample complexity bounds for $\ell_p$ distances up to $\operatorname{polylog}\left(\frac{1}{\epsilon}, \frac{1}{\delta}\right)$

*Proof.* For the first two cases, we assume $d < n$, since otherwise, the bounds follow from Theorem 2.7. Our proof uses the following result of Warren (1968) from algebraic geometry:

**Fact 3.3** (Warren (1968))**.** *Let $m \geq \ell \geq 2$ be integers, and let $P_1, \ldots, P_m$ be real polynomials on $\ell$ variables, each of degree $\leq k$. Let*

$$U(P_1, \ldots, P_m) = \left\{ \vec{x} \in \mathbb{R}^\ell \mid P_i(\vec{x}) \neq 0 \, for \, all \, i \in [m] \right\}$$

*be the set of points $x \in \mathbb{R}^\ell$ which are non-zero in all polynomials. Then the number of connected components in $U(P_1, \ldots, P_m)$ is at most $(4ekm/\ell)^\ell$.*

In order to bound the sample complexity of the contrastive learning problem, by Lemma 2.6 it suffices to bound its Natarajan dimension, which in the binary case is equivalent to the VC dimension. In particular, we show that for a dataset $V$ of size $n$, the VC-dimension of contrastive learning for $\ell_p$-distances in dimension $d$ is $O(n \min(d, n))$ for even $p \geq 2$ and $O(nd \log n)$ for odd $p \geq 1$. For this it suffices to show that for every set of queries $Q = \{(u_i, v_i, w_i)\}_{i=1}^m$ of size $m = \tilde{\Omega}(n \min(d, n))$, there exists labeling of $Q$ that cannot be satisfied by any embedding in a $d$-dimensional $\ell_p$-space. We split the analysis into cases for even $p$ and odd $p$.

**Upper Bound for Even $p$.** We associate each data point $v \in V$ with a set of $d$ variables $x_1(v), \ldots, x_d(v)$, corresponding to coordinates of $v$ after embedding in the $d$-dimensional space. Let $\vec{x} = (x_1(v_1), \ldots, x_d(v_1), \ldots, x_1(v_n), \ldots, x_d(v_n))$. For every constraint $(u, v, w)$, we define the polynomial $P_{u,v,w} \colon \mathbb{R}^{nd} \to \mathbb{R}$ as

$$P_{(u,v,w)}(\vec{x}) = \sum_{j=1}^d (x_j(u) - x_j(v))^p - \sum_{j=1}^d (x_j(u) - x_j(w))^p.$$

Note that $P_{(u,v,w)}(\vec{x}) < 0$ if and only if the constraint $(u, v^+, w^-)$ is satisfied, and $P_{(u,v,w)}(\vec{x}) > 0$ if and only if the constraint $(u, w^+, v^-)$ is satisfied. Finally, we define $P_1, \ldots, P_m$ for $i \in [m]$ as $P_i(\vec{x}) = P_{u_i,v_i,w_i}(\vec{x})$.

For any labeling $\vec{Q}$ of the queries $Q$ and for any $i \in [m]$, we define $s_i(\vec{Q}) = 1$ if the $i$'th query $(u_i, v_i, w_i) \in Q$ is labeled as $(u_i, v_i^+, w_i^-)$, and $s_i(\vec{Q}) = -1$ otherwise.

We make the following observations, which are proven in Section B:

**Lemma 3.4.** *For $s \in \{-1, 1\}^m$, define $C_s = \{\vec{x} \in \mathbb{R}^d \mid \operatorname{sign} P_i(\vec{x}) = s_i \, \text{for all } i\}$. Then:*

1. *For distinct $s, s' \in \{-1, 1\}^m$ we have $C_s \cap C_{s'} = \emptyset$.*
2. *Each $C_s$ is either empty or is a union of connected components of $U(P_1, \ldots, P_m)$.*
3. *Let $\vec{Q}$ be a labeling of $Q$. Then $C_{s(\vec{Q})} \neq \emptyset$ if and only if there is a mapping $V \to \mathbb{R}^d$ satisfying all the distance constraints of $\vec{Q}$.*

We now complete the proof for the case of even $p$. Defining $P_1, \ldots, P_m$ as above, by Theorem 3.3, there are at most $(4epm/nd)^{nd}$ connected components in the set $U(P_1, \ldots, P_m)$. Since for two labelings $\vec{Q}_1, \vec{Q}_2$ of $Q$ it holds that $s(\vec{Q}_1) \neq s(\vec{Q}_2)$, then by Lemma 3.4 either the sets $C_{s(\vec{Q}_1)}$ and $C_{s(\vec{Q}_2)}$ are different connected components, or at least one of them is empty. Moreover, the number of possible labelings of $Q$ is $2^m$, which is greater than $(4epm/nd)^{nd}$ when choosing $m \geq cnd$ for a sufficiently large constant $c$. Therefore, for at least one labeling $\vec{Q}$ it holds that $C_{s(\vec{Q})} = \emptyset$. Since there is no embedding satisfying the distance constraints of $\vec{Q}$, the claim follows.

**Upper bound for odd $p$.** In the case of odd $p$ it suffices to show that for a dataset of size $n$ in dimension $d$, the VC-dimension of contrastive learning for $\ell_p$-distances is $O(nd \log n)$. Unlike in the even $p$ case, our distance constraints are comprised of sums of functions of the form $|x_j(u) - x_j(v)|^p$, and are thus not polynomial constraints. On the other hand, we note that if we have some fixed ordering of the points w.r.t. each coordinate, these constraints become polynomials. To address the issue, we enumerate all possible choices for the ordering of the points w.r.t. each coordinate and show that there is a labeling $\vec{Q}$ such that no embedding satisfies it, regardless of the ordering.

Similarly to the case of even $p$, we associate with each $v \in V$ a set of $d$ variables $x_1(v), \ldots, x_d(v)$. Let $\vec{x} = (x_1(v_1), \ldots, x_d(v_1), \ldots, x_1(v_n), \ldots, x_d(v_n))$. For every coordinate $i \in [d]$, we fix the

ordering of the points w.r.t. this coordinate, i.e. we fix a permutation $\pi^{(i)} \colon [n] \to [n]$ such that $x_i(v_{\pi^{(i)}(1)}) \leq x_i(v_{\pi^{(i)}(2)}) \leq \cdots \leq x_i(v_{\pi^{(i)}(n)})$, and bound the number of satisfiable labelings which respect this ordering. We define $\sigma_{uv}^{(i)} = 1$ if $x_i(u) \geq x_i(v)$ and $\sigma_{uv}^{(i)} = -1$ otherwise, and refer to $\sigma$ as the *order*. Then, for any constraint $(u, v, w)$, define the polynomial $P_{u,v,w} \colon \mathbb{R}^{nd} \to \mathbb{R}$:

$$P_{(u,v,w)}(\vec{x}) = \sum_{j=1}^{d} \sigma_{uv}^{(j)}(x_j(u) - x_j(v))^p - \sum_{j=1}^{d} \sigma_{uw}^{(j)}(x_j(u) - x_j(w))^p.$$

Note that $P_{(u,v,w)}(\vec{x}) < 0$ if and only if the constraint $(u, v^+, w^-)$ is satisfied for the selected order $\pi^{(1)}, \ldots, \pi^{(d)}$ (and likewise for $P_{(u,v,w)}(\vec{x}) > 0$ and $(u, w^+, v^-)$). Similarly to the case of even $p$, we define $P_1, \ldots, P_m$ for $i \in [m]$ as $P_i(\vec{x}) = P_{u_i, v_i, w_i}(\vec{x})$, and for any labeling $\vec{Q}$ of queries $Q$, for all $i \in [m]$, we define $s_i(\vec{Q}) = 1$ if the $i$'th query $(u_i, v_i, w_i) \in Q$ is labeled as $(u_i, v_i^+, w_i^-)$, and $s_i(\vec{Q}) = -1$ otherwise.

As in the case of even $p$, by Fact 3.3, there are at most $(4epm/nd)^{nd}$ connected components in the set $U(P_1, \ldots, P_m)$. Therefore, there are at most $(4epm/nd)^{nd}$ choices of labels which are satisfiable in a manner that respects the ordering $\pi^{(1)}, \ldots, \pi^{(d)}$.

Taking $m = \Omega(nd \log n)$, we have that $\frac{2^m}{n^{nd}} > \left(\frac{4epm}{nd}\right)^{nd}$. Since there are $(n!)^d < n^{nd}$ possible choices of the permutations $\pi^{(1)}, \ldots, \pi^{(n)}$ and hence at most as many orders $\sigma$, there are at most $(4epm/nd)^{nd} \cdot n^{nd}$ possible labelings for which there exists some order such that the set of constraints is satisfiable and respects this order. Since $(4epm/nd)^{nd} \cdot n^{nd}$ is less than $2^m$, there exists a choice of a labeling $\vec{Q}$ for which the constraints are not satisfiable for any order, i.e. $Q$ is not shattered.

**Upper Bound for Constant $d$.** We will proof the following statement: for constant odd $p \in (0, +\infty)$, a dataset $V$ of size $n$, and the $\ell_p$ distance $\rho_p \colon V \times V \to \mathbb{R}$ in a $d$-dimensional space, the VC dimension of contrastive learning is $O(nd^2)$. This gives optimal bound for constant $d$.

Similarly to the proof above, we assume that $d^2 < n$. We then prove the sample complexity bound by bounding the VC dimension of the problem by $O(nd^2)$, and obtain the sample bounds using Theorem 2.6.

Similar to previous cases, we associate with each $v \in V$ a set of $d$ variables $x_1(v), \ldots, x_d(v)$. Let $\vec{x} = (x_1(v_1), \ldots, x_d(v_1), \ldots, x_1(v_n), \ldots, x_d(v_n))$.

We associate each constraint with $2^{2d}$ polynomials in the following manner: for any constraint $(u, v, w)$ and any choice $\tau \in \{-1, 1\}^{2d}$, we define the polynomial $P_{\tau,u,v,w} \colon \mathbb{R}^{\ell} \to \mathbb{R}$ as

$$P_{\tau,u,v,w}(\vec{x}) = \sum_{j=1}^{d} \tau(j)(x_j(u) - x_j(v))^p - \sum_{j=1}^{d} \tau(d+j)(x_j(u) - x_j(w))^p,$$

where $\tau(j)$ is the value of the $j$'th coordinate of $\tau$.

We note that the sign pattern of the set of polynomials $\{P_{\tau,u,v,w}(\vec{x}) \gtrless 0\}_{\tau \in \{0,1\}^{2d}}$ determines the sign of $\sum_{j=1}^{d} |x_j(u) - x_j(v)|^p - \sum_{j=1}^{d} |x_j(u) - x_j(w)|^p$. Therefore, for different choices of signs for the constraints, the solutions satisfying them must be contained in different connected components of $U(P_1, \ldots, P_{2^{2d} \cdot m})$.

By Fact 3.3, there are at most $(4ep2^{2d}m/nd)^{nd}$ connected components in the set $U(P_1, \ldots, P_{2^{2d} \cdot m})$. Therefore, taking $m \geq cnd^2$ for sufficiently large $c$ (depending on $p$), we get that $2^m > (4ep2^{2d}m/nd)^{nd}$, meaning there is a choice of signs for which there is no solution, i.e. these samples cannot be shattered.

$\square$

## 4 EXTENSIONS

**Reducing dependence on $n$** While our results show that the dependence on $n$ is necessary, in practice it can be avoided by making additional assumptions. An extremely popular assumption is

that the existence of $k \ll n$ latent classes in the data (see e.g. Saunshi et al. (2019)). In this case, one can consider using an unsupervised clustering algorithm (e.g. a pretrained neural network) to partition the points into $k$ clusters and then apply our results to classes instead of individual points, effectively replacing $n$ with $k$ in all our bounds.

**Extension to $k$ negative samples:** In this extension, we consider a setting in which each sample is a tuple $(x, x_1, \ldots, x_{k+1}) \in V^{k+2}$, and a labeling $(x, x_i^+, x_1^-, \ldots, x_{i-1}^-, x_{i+1}^-, \ldots, x_{k+1}^-)$ is a choice of an example $x_i$ which minimizes $\min_{j \in [k+1]} \rho(x, x_j)$, i.e. $\rho(x, x_i) < \rho(x, x_j)$ for all $j \neq i$. As an immediate corollary to our results, we obtain a bound on the sample complexity for the $k$ negative setting (in fact, our results for all distance functions considered in the paper extend to this setting, see proof in Appendix C.1):

**Theorem 4.1.** *For a constant p, a dataset $V$ of size $n$, and the $\ell_p$ distance $\rho_p \colon V \times V \to \mathbb{R}$ in a d-dimensional space, the sample complexity of contrastive learning $\ell_p$-distances with k negatives is bounded as shown in Table 3*

| $p$ | Realizable? | Upper bound up to polylog $\left(\frac{1}{\epsilon}, \frac{1}{\delta}\right)$ | Lower bound up to polylog $\left(\frac{1}{\epsilon}, \frac{1}{\delta}\right)$ |
|---|---|---|---|
| Even | Realizable | $\min(nd, n^2) \log(k+1)/\epsilon$ | $\min(nd, n^2)/\epsilon$ |
| Even | Agnostic | $\min(nd, n^2) \log(k+1)/\epsilon^2$ | $\min(nd, n^2)/\epsilon^2$ |
| Odd | Realizable | $\min(nd \log n, n^2) \log(k+1)/\epsilon$ | $\min(nd, n^2)/\epsilon$ |
| Odd | Agnostic | $\min(nd \log n, n^2) \log(k+1)/\epsilon^2$ | $\min(nd, n^2)/\epsilon^2$ |

Table 3: Bounds on sample complexity for $\ell_p$ distances with $k$ negatives

**Well-separated samples:** In this extension, we focus on the most popular $\ell_2$-distance and consider an "approximate" setting, where the positive and negative samples are guaranteed to be well-separated, i.e. their distance from the anchor differs by at least a multiplicative factor. This scenario is highly motivated in practice since the positive example is typically sampled to be much closer to the anchor than the negative example. We show that in this setting the sample complexity can be improved to be almost independent of $d$.[7]

For a parameter $0 < \alpha < 1$, we assume that our sample distribution $\mathcal{D}$ has the following property: If $(x, y^+, z^-) \in \text{supp}(\mathcal{D})$, then $\rho(x, z) > (1 + \alpha)\rho(x, y)$, or $\rho(x, y) > (1 + \alpha)\rho(x, z)$. We call a distribution with this property *well-separated*. We show that for the $\ell_2$-distance, the sample complexity of the problem in this setting is between $\widetilde{\Omega}_{\epsilon,\delta}(n/\alpha)$ and $\widetilde{O}_{\epsilon,\delta}(n/\alpha^2)$. This result is proven in Appendix E.

**Theorem 4.2.** *The sample complexity of $(1 + \alpha)$-separate contrastive learning for $\ell_2$ distance is $S_3(\epsilon, \delta) = \widetilde{O}_{\epsilon,\delta}\left(\frac{n}{\alpha^2}\right)$ and $S_3(\epsilon, \delta) = \widetilde{\Omega}_{\epsilon,\delta}\left(\frac{n}{\alpha}\right)$.*

**Contrastive Learning for Other Distance Functions:** In Appendix D, we show near-tight sample complexity bounds for contrastive learning in other distance functions, such as tree metrics, cosine similarity, class distances, and prove tight general bounds for arbitrary metrics. We also extend our results to quadruplet contrastive learning.

## 5 Conclusion

In this paper we have given a theoretical analysis of contrastive learning, focusing on the generalization in the PAC-learning setting. Our results show (in most cases) tight dependence of the sample complexity on the dimension of the underlying representation. It remains open whether our results can be made optimal for $\ell_p$-distances for odd $p$. It is also open how the results can be improved via a suitable choice of batched adaptively chosen comparisons (see e.g. Zou et al. (2015)). Note that in the non-batched adaptive query setting optimal bounds on the query complexity are known to be $\Theta(n \log n)$ (Kannan et al., 1996; Emamjomeh-Zadeh & Kempe, 2018). On the experimental side, while we show asymptotic convergence between our theory and practice, it might be worth looking into further refinements of our approach which can better capture the fact that the constant ratio between the observed and predicted values tends to be small.

---

[7]Disregarding polylog factors in $d$.

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

## A  EXAMPLE

**Example and notation**  We are given 4 data points: images of a cat, a rat, a plane, and a train, which we denote as $V = \{c, r, p, t\}$. We consider the realizable case, i.e. there exists a ground truth metric $\rho$ on these points (e.g. general or $\ell_p$) with error rate $\epsilon^* = 0$. For this example, we will assume that we embed these points into $\mathbb{R}$, i.e. $d = 1$. Hence the hypothesis class $\mathcal{H}$ is just a class of all Euclidean metrics on $\mathbb{R}$.

For these data points, there are 12 possible input samples, corresponding to 4 choices of an anchor and 3 choices of the two other query points. For this example, let $\mathcal{D}$ be the distribution of samples $(x, y^+, z^-)$ such that each input $(x, y, z)$ is sampled uniformly, and the label is decided according to which of $y$ and $z$ is closer to $x$ according to $\rho$. For example, if the input is $(c, r, p)$, we expect the ground truth $\rho$ to be that the cat is closer to the rat rather than to the plane, which we write as $(c, r^+, p^-)$. Hence, this version of the problem is just a binary classification problem.

**Error rate**  If, for example, the ground-truth embedding $\rho$ was $r \to 0, c \to 1, p \to 3, t \to 10$, then, assuming the distribution on the inputs is uniform, we achieve the error rate $\epsilon = \frac{1}{6}$, since only 2 of 12 queries - $(p, c^+, r^-)$ and $(p, c^+, r^-)$ - are not satisfied.

**Sampling mechanism and learning scheme**  Since we have a binary classification problem, without a training set, we can only predict randomly, which gives the error rate $\epsilon = \frac{1}{2}$. Now, suppose that we have a single training example $(c, r^+, p^-)$ sampled from $\mathcal{D}$. The algorithm that achieves the best worst-case guarantee is an empirical risk minimizer (ERM), which seeks to find the embedding satisfying the maximum number of constraints. In this case, the constraint is satisfiable e.g. by embedding $c \to 0, r \to 1, p \to 3, t \to 10$, since $|c - r| = 1 < |c - p| = 3$.

**Example of Shattering**  As an example, the set $S = \{(c, r, p), (c, p, t), (c, r, t)\}$ is not shattered since for the labeling $\{(c, r^+, p^-), (c, p^+, t^-), (c, t^+, r^-)\}$ there exists no embedding such that $\rho(c, r) < \rho(c, p), \rho(c, p) < \rho(c, t)$, and $\rho(c, r) > \rho(c, t)$. On the other hand, the set $S' = \{(c, p, t), (r, p, t)\}$ is shattered: a labeling of $S'$ determines for each of $c, r$ which point is closer between $p$ and $t$. For any labeling of $S'$, we can embed $p, t$ arbitrarily and embed each of $c, r$ closer to either $p$ or $t$ according to the labeling, thus satisfying the constraints imposed by the labeling.

## B  MISSING PROOFS FROM SECTION 3

*Proof of Lemma 3.4.*

1. If $\vec{x} \in C_s$ and $\vec{y} \in C_{s'}$ for and $s$ and $s'$ which differ in index $j$, we know that sign $P_j(\vec{x}) \neq$ sign $P_j(\vec{y})$, and hence $\vec{x} \neq \vec{y}$. Hence, $C_s$ and $C_{s'}$ are disjoint.

2. First note that all $C_s$ are open since polynomials are continuous functions, and preimages of open sets $(-\infty, 0)$ and $(0, +\infty)$ are open. Moreover, $\cup_s C_s = U(P_1, \ldots, P_m)$ and all $C_s$ are disjoint, and hence $\{C_s\}_s$ is a partition of $U(P_1, \ldots, P_m)$. By definition of a connected set, it's impossible to partition a connected component of $U(P_1, \ldots, P_m)$ into multiple open sets, and hence each $C_s$ must be a union of some connected components of $U(P_1, \ldots, P_m)$.

3. Follows from our choice of polynomials: $P_{(u,v,w)}(\vec{x}) > 0$ if $\vec{x}(u)$ is closer to $\vec{x}(v)$ than to $\vec{x}(w)$, and $P_{(u,v,w)}(\vec{x}) < 0$ otherwise.  □

### B.1  MISSING LOWER BOUND OF SECTION 3

Recall that our goal is to learn a distance function $\rho : V \times V \to \mathbb{R}$ which can be produced by some allocation of points in the $d$-dimensional Euclidean space. In other words, the set of labeled queries $\{(x_i, y_i, z_i)\}_i$ is realizable if there exists a mapping $f : V \to \mathbb{R}^d$ such that $\|f(x_i) - f(y_i)\|_2 < \|f(x_i) - f(z_i)\|_2$.

**Theorem B.1.** *For a dataset of size $n$, the VC-dimension of contrastive learning for $\ell_p$-distances for any $p \in (0, +\infty)$ in dimension $d$ is $\Omega(\min(n^2, nd))$*

*Proof.* We first assume that $d < n$, since otherwise the bound follows from Theorem 2.7. Let $V$ be the set of points of size $n$. We will present a set of queries $S$ of size $\Omega(nd)$ that can be shattered. Recall that it means that for any $T \subseteq S$ there exists a mapping $f \colon V \to \mathbb{R}^d$, such that:

- triplets from $T$ are satisfied, i.e. each triplet $(x, y, z) \in T$ is labeled $(x, y^+, z^-)$, meaning $\rho(x, y) < \rho(x, z)$;

- triplets from $S \setminus T$ are not satisfied, i.e. each triplet $(x, y, z) \in S \setminus T$ is labeled $(x, z^+, y^-)$, meaning $\rho(x, y) > \rho(x, z)$.

First, note that for $d = 1$, any set of $\lfloor n/3 \rfloor$ disjoint queries gives $\Omega(n)$ lower bound. For the rest of the proof, we assume that $d > 1$.

We partition $V$ arbitrarily into disjoint sets $A$ and $B$ of sizes $n - d$ and $d$ respectively. Intuitively, the points from $A$ always act like anchors, and points from $B$ never act like anchors.

We first describe how to map $B = \{v_1, \ldots, v_d\}$ (their mapping won't depend on the labels of the queries). For each $i \in [d]$, we map $f(v_i) = e_i$, where $e_i$ is the $i$'th standard basis vector.

Next, we describe the queries. As defined above, $A = \{x_1, \ldots, x_{n-d}\}$ is a set of anchors, and for each anchor $x_i$, the queries are of form $(x_i, v_1, v_2), (x_i, v_1, v_3), \ldots, (x_i, v_1, v_d)$. Hence, there exist $(d-1)(n-d)$ queries, as required by the theorem, and it remains to show that this set of queries can be shattered. Clearly, since the arrangement of points $v_1, \ldots, v_d$ is fixed, allocation of anchor $x_i$ doesn't affect queries with another anchor $x_j$, and hence it suffices to map every anchor independently from each other.

Let's fix the anchor $x_i$. Let the first coordinate of $f(x_i)$ be $1/2$. For $j \in [2 : d]$, if $(x_i, v_1, v_j)$ is labeled $(x_i, v_1^+, v_j^-)$, then we select the the $j$-th coordinate of $f(x_i)$ to be 0, and otherwise we select it to be 1. Note that then the query is satisfied: the summations for $\|f(x_i) - e_1\|_p^p$ and $\|f(x_i) - e_j\|_p^p$ differ only in the first and the $j$-th coordinates. When the $j$-th coordinate is 0, we have

$$\|f(x_i) - e_1\|_p^p - \|f(x_i) - e_j\|_p^p = (1/2)^p - ((1/2)^p + 1^p) < 0,$$

and hence $(x_i, v_1^+, v_j^-)$ is satisfied. On the other hand, when the $j$-th coordinate is 1, we have

$$\|f(x_i) - e_1\|_p^p - \|f(x_i) - e_j\|_p^p = ((1/2)^p + 1^p) - (1/2)^p > 0,$$

and hence $(x_i, v_j^+, v_1^-)$ is satisfied. Hence, we can construct $f(x_i)$ that satisfies all the queries with anchor $x_i$. Therefore, we can shatter the set of all $(d-1)(n-d)$ queries, finishing the proof. $\square$

## C  VARIATIONS

### C.1  CONTRASTIVE LEARNING WITH $k$ NEGATIVES

In the previous sections, we considered the queries of the form (anchor, positive, negative). In contrastive learning, it's common to have multiple negative examples. In this section, we show the following result which can be used to derive bounds for the contrastive learning with $k$ negatives.

**Theorem C.1.** *Let $\Delta$ be the class of allowed metric functions. Let $\mathrm{Ndim}$ be the VC dimension of the contrastive learning problem on $\Delta$ with 1 negative. Then the Natarajan dimension for the contrastive learning problem on $\Delta$ with $k$ negatives is at most $\mathrm{Ndim}$.*

*Assume additionally that for any set of points $S$ and any distance function $\rho \in \Delta$ on $S$, for any point $o$ there exists a distance function $\rho' \in \Delta$ on $S \cup \{o\}$ such that $\rho'|_{S \times S} = \rho$ and $\rho(x, y) < \rho(x, o)$ for any $x, y \in S$ [8]. Then, for the dataset on $n$ points such that $n - k = \Omega(n)$, the Natarajan dimension of contrastive learning on $\Delta$ with $k$ negatives is $\Omega(\mathrm{Ndim})$.*

---

[8]Intuitively, this condition implies that for any set $S$ of points we can find an "outlier" – a point $o$ which is sufficiently far from the existing set of points. This assumption is naturally satisfied for $\ell_p$ and tree distances.

*Proof.* For the upper bound, assume that for some query $(x, x_1, \ldots, x_{k+1})$, for the Natarajan shattering we select the $x_i$ and $x_j$ for some $i$ and $j$, i.e. the possible labels are $(x, x_i^+, x_1^-, \ldots, x_{i-1}^-, x_{i+1}^-, \ldots, x_{k+1}^-)$ and $(x, x_j^+, x_1^-, \ldots, x_{j-1}^-, x_{j+1}^-, \ldots, x_{k+1}^-)$. Then, the first query implies $(x, x_i^+, x_j^-)$, and the second query implies $(x, x_j^+, x_i^-)$. Hence, any Natarajan-shattered set with $k$ negatives corresponds to a VC-shattered set of queries with 1 negatives of the same cardinality. Hence, the Natarajan dimension is upper-bounded by the VC dimension of the 1-negative problem.

For the lower bound, we split the set of points $V$ into two sets: "outlier" points $O = \{o_1, \ldots, o_{k-1}\}$ and the remaining points $S = \{v_1, \ldots, v_{n-k+1}\}$. Let $Q = \{(x_i, y_i, z_i)\}_i$ be the VC-shattered set for the 1-negative contrastive learning on $n - k + 1$ points. Then, the Natarajan-shattered set of queries for the $k$-negative contrastive learning on $n$ points are $Q' = \{(x_i, y_i, z_i, o_1, \ldots, o_{k-1})\}_i$. For each query, for the Natarajan shattering we select $y_i$ and $z_i$, meaning that possible labels are $(x_i, y_i^+, z_i^-, o_1^-, \ldots, o_{k-1}^-)\}_i$ and $(x_i, z_i^+, y_i^-, o_1^-, \ldots, o_{k-1}^-)\}_i$

Since the set of $Q$ is shattered, for any choice of labeling of $\{(x_i, y_i, z_i)\}_i$ there exists a distance function $\rho \in \Delta$ satisfying this labeling. It suffices to guarantee that $\rho(x_i, y_i) < \rho(x_i, o_j)$ and $\rho(x_i, z_i) < \rho(x_i, o_j)$ for all $i$ and $j$, and by our assumption, there exists a distance function satisfying this condition. □

**Corollary C.2.** *The same upper bounds on sample complexities for all distance functions considered in the paper (i.e. in Theorem 3.2, Theorem 2.7, Theorem D.2, and Theorem D.4) hold for $k$-negative contrastive learning $S_k(\epsilon, \delta)$ and $S_k^a(\epsilon, \delta)$ up to the $\log k$-factor. The same lower bounds that hold for contrastive learning in $\ell_p$-norm and tree metrics (i.e. Theorem 3.1 and Theorem D.2) also hold for $k$-negative contrastive learning.*

## C.2 LEARNING ON QUADRUPLETS

Recall that we are given a set of quadruplets $\{((x^+, y^+), (z^-, w^-))\}$, where each quadruplet $((x^+, y^+), (z^-, w^-))$ imposes constraint $\rho(x^+, y^+) < \rho(z^-, w^-)$.

**Theorem C.3.** *For the VC dimension of contrastive learning on quadruplets, we have the same bounds as for the learning on triplets. Namely, for a dataset of size $n$, the following holds.*

- *The VC dimension for arbitrary metric is $\Theta(n^2)$.*

- *The VC dimension for $\ell_p$-distances in dimension $d$ is $\Omega(nd)$ for all $p \in (0, \infty)$.*

- *The VC dimension for $\ell_p$-distances in dimension $d$ is $O(nd)$ for even $p$ and $O(nd \log n)$ for odd $p$.*

- *The VC dimension for the tree metric is $\Omega(n) \cap O(n \log n)$.*

*Proof.* First, we note that any lower bound in the triplet case is also a lower bound for the quadruplet case, since a triplet query $(x, y, z)$ is equivalent to the quadruplet query $((x, y), (x, z))$.

For the upper bounds, for $\ell_p$ and tree metric, we use the same approach: each constraint $((x, y), (z, w))$ corresponds to a polynomial (potentially after fixing the order of points or the tree structure). Since the number of variables and polynomials doesn't change, we get the same upper bounds. Finally, for arbitrary metric, similarly to Theorem 2.7, for each set of queries, we can construct a graph with vertices from $V \times V$: for each query $((x, y), (z, w))$, we create an undirected edge, and labeling the query is equivalent to orienting the edge. If there is a cycle, it's possible to get a contradiction, and hence we can't have more than $O(|V \times V|) = O(n^2)$ queries. □

## D CONTRASTIVE LEARNING IN OTHER METRICS

In this section, we discuss related results. First, we show a bound for arbitrary distance functions:

**Theorem D.1** (Arbitrary distance)**.** *For an arbitrary distance function $\rho \colon V \times V \to \mathbb{R}$ and a dataset of size $n$, the sample complexity of contrastive learning is $S_3(\epsilon, \delta) = \Theta\left(\frac{n^2}{\epsilon} \operatorname{polylog}\left(\frac{1}{\epsilon}, \frac{1}{\delta}\right)\right)$ in the*

*realizable case and* $S_3^a(\epsilon, \delta) = \Theta\left(\frac{n^2}{\epsilon^2} \text{polylog}\left(\frac{1}{\epsilon}, \frac{1}{\delta}\right)\right)$ *in the agnostic case. Furthermore, the lower bounds hold even if* $\rho$ *is assumed to be a metric.*

*Proof.* We prove this by showing a $\Theta(n^2)$ bound on the VC dimension of contrastive learning with arbitrary distance functions.

**Upper bound.** Consider any set of samples $\{(x_i, y_i, z_i))\}_{i=1...k}$ of size $k \geq n^2$. There exists $x$ such that there are at least $n$ samples which have $x$ as their first element. We denote these samples as $(x, y_{i_1}, z_{i_1}), \ldots, (x, y_{i_n}, z_{i_n})$. Consider a graph that has a vertex corresponding to each element in the dataset. Create an undirected edge in this graph between the $n$ pairs of vertices $(y_{i_1}, z_{i_1}), \ldots, (y_{i_n}, z_{i_n})$.

Since the number of edges is equal to the number of vertices, there must exist a cycle $\mathcal{C}$ in this graph. We can index the vertices along this cycle as $v_1, v_2, \ldots, v_t$. Now consider the following labeling of the samples:

$$\rho(x, v_1) < \rho(x, v_2) < \rho(x, v_3) < \cdots < \rho(x, v_t) < \rho(x, v_1).$$

This labeling is inconsistent with any distance function and hence not all different labelings of this sample are possible. Since the same argument applies for any sample of size at least $n^2$, an $n^2$ upper bound on the VC-dimension follows.

**Lower Bound.** Let $V = \{v_1, \ldots, v_n\}$. We prove that the set of queries

$$Q = \cup_{i \in [n]}\{(v_i, v_{i+1}, v_{i+2}), (v_i, v_{i+2}, v_{i+3}), \ldots, (v_i, v_{n-1}, v_n)\}$$

is shattered. Let $\vec{Q}$ be a labeling of $Q$. For every $i \in [n]$, we define a graph $H_i = (\{v_{i+1}, \ldots, v_n\}, E_i)$ where $E_i$ contains a directed edge $(v_j, v_{j+1})$ for each query, $(v_i, v_j, v_{j+1})$, orientated towards the negative example according to $\vec{Q}$. The graph $H_i$ is acyclic, as it is an orientation of a path. Therefore we can topologically sort $H_i$, and obtain some topological order $p_{i+1}^{(i)}, \ldots, p_n^{(i)}$ on the vertices $v_{i+1}, \ldots, v_n$.

Consider a metric where the distance between two data items $v_i, v_j$ is defined as $\rho(v_i, v_j) := n + p_j^{(i)}$ for each $i < j \leq n$. We note that this is indeed a metric: triangle inequalities are satisfied since all distances are in the range $[n, 2n]$. Finally, we note this distance function satisfies all the queries. Indeed, for $i, j, k$ such that $i < k, j$ and $|k - j| = 1$, if $(v_i, v_j^+, v_k^-) \in \vec{Q}$ then the directed edge $(v_j, v_k) \in E_i$, therefore $\rho(v_i, v_k) > \rho(v_i, v_j)$ since $p_k^{(i)} > p_j^{(i)}$. $\qquad\square$

A metric $(V, \rho)$ is called a *tree metric* if there exists a tree $T$ with weighted edges and $n$ leaves, such that for each $v \in V$ there is a unique leaf $l(v)$ associated with it, and such that for each $u, v \in V$, $\rho(u, v)$ is equal to the sum of weights along the unique path between $l(u), l(v)$ in $T$.

**Theorem D.2** (Tree distance). *For the tree metric distance function* $\rho_T : V \times V \to \mathbb{R}$, *corresponding to distances between the nodes of a tree with leaves from* $V$, *the sample complexity of contrastive learning in the realizable case is* $S_3(\epsilon, \delta) = O\left(\frac{n \log n}{\epsilon} \text{polylog}\left(\frac{1}{\epsilon}, \frac{1}{\delta}\right)\right)$ *and* $\Omega\left(\frac{n}{\epsilon} \text{polylog}\left(\frac{1}{\epsilon}, \frac{1}{\delta}\right)\right)$, *and in the agnostic case is* $S_3^a(\epsilon, \delta) = O\left(\frac{n \log n}{\epsilon^2} \text{polylog}\left(\frac{1}{\epsilon}, \frac{1}{\delta}\right)\right)$ *and* $\Omega\left(\frac{n}{\epsilon^2} \text{polylog}\left(\frac{1}{\epsilon}, \frac{1}{\delta}\right)\right)$.

*Proof.* We prove this by showing that the VC-dimension of contrastive learning for the tree metric is $O(n \log n)$. First, we may assume that the number of vertices in the tree is at most $2n$. Indeed, any induced path in the tree can be contracted to a single edge whose weight is the sum of weights along the path. After this procedure, the tree metric remains the same and the resulting tree has no vertices of degree 2. It follows by a counting argument that the number of vertices in this tree is at most $2n$. The rest of the proof is similar to the odd case of the Theorem 3.2: we enumerate over all possible tree structures with at most $2n$ vertices and allocation of the data items onto the tree vertices. By Cayley's formula (Cayley, 1878), there exists $2(2n)^{2n-2}$ such different tree structures and vertex allocations. For a fixed tree, the distance between every pair of vertices can be expressed as a linear combination of the edge weights. Hence, every constraint becomes a linear inequality on the edge weights.

The rest of the proof is analogous to that of Theorem 3.2. Fix a tree $T$. Let $E_T(u,v)$ be the set of edges in the path between $u,v$ in $T$. We have $k \le 2n - 1$ variables $\vec{x} = (x_1, \ldots, x_k)$, where $x_i$ represents the edge weight of $e_i$. For $m$ queries we define polynomials $P_1(\vec{x}), \ldots, P_m(\vec{x})$, one for each query, such that for query $(u_i, v_i, w_i)$ we define the polynomial

$$P_i(\vec{x}) = \sum_{e_i \in E_T(u_i,v_i)} x_i - \sum_{e_i \in E_T(u_i,w_i)} x_i.$$

I.e. $(u_i, v_i^+, w_i^-)$ is satisfied if and only $P_i(\vec{e}) < 0$. For $m = 3n \log n$ queries, by Theorem 3.3 the number of possible sign combinations is at most $(4em/k)^k < \frac{2^m}{2(2n)^{2n-2}}$. Summing over all possible tree structures, the number of sign combinations is less than $2^m$, and hence the set of $m$ queries can't be shattered. $\square$

The Cosine Similarity function $\cos : \mathbb{R}^d \times \mathbb{R}^d \to \mathbb{R}$ is a function which returns for each pair of points $x, y$ the cosine of their angle, i.e. $\cos \angle(x,y)$.

**Theorem D.3** (Cosine Similarity). *For the cosine similarity function, the sample complexity of contrastive learning $S_3(\epsilon, \delta) = \Theta\left(\frac{\min(n^2, nd)}{\epsilon} \operatorname{polylog}\left(\frac{1}{\epsilon}, \frac{1}{\delta}\right)\right)$ in the realizable case and is $S_3^a(\epsilon, \delta) = \Theta\left(\frac{\min(n^2, nd)}{\epsilon^2} \operatorname{polylog}\left(\frac{1}{\epsilon}, \frac{1}{\delta}\right)\right)$ in the agnostic case.*

*Proof.* As usual, we assume that $d < n$, since otherwise the result follows from Theorem 2.7. We prove this by showing that the VC-dimension of contrastive learning for cosine similarity in $\mathbb{R}^d$ is $\Theta(nd)$. Recall that $\cos \angle(x,y) = \frac{\langle x,y \rangle}{\|x\|_2 \cdot \|y\|_2} = \langle \frac{x}{\|x\|_2}, \frac{y}{\|y\|_2} \rangle$, and hence w.l.o.g. we can assume that all points are unit vectors. For unit vectors $x, y$ we have $\|x - y\|_2^2 = \|x\|_2^2 + \|y\|_2^2 - 2\langle x,y \rangle = 2 - 2\cos \angle(x,y)$, and hence $\cos \angle(x, y^+) > \cos \angle(x, z^-)$ is equivalent to $\|x - y^+\|_2 < \|x - z^-\|_2$. To summarize, contrastive learning for cosine similarity is equivalent to contrastive learning for $\ell_2$ distance of points on the sphere.

The upper bound $O(nd)$ for VC dimension follows directly from the fact that for the cosine similarity we only consider unit vectors, and hence the hypothesis space is less than that for the $\ell_2$-distance (which allows arbitrarily-normed vectors).

For the lower bound, we repeat the proof of Theorem B.1, with minor alterations that ensure that all points are embeddable in the unit sphere.

Similar to Theorem B.1, we partition $V$ into two sets $A = \{x_1, \ldots, x_{n-d}\}$ as a set of anchors, and $B = \{v_1, \ldots, v_d\}$. We define the query set $Q$ as follows: for each anchor $x_i$, the set $Q$ contains the queries $(x_i, v_1, v_2), (x_i, v_1, v_3), \ldots, (x_i, v_1, v_d)$.

We set $f(v_i) = e_i$, i.e. we embed $v_i$ to the $i$'th standard vector $e_i$.

For each $x_i$, we define a vector $g(x_i) \in \mathbb{R}^d$ in the following manner. Let the first coordinate of $g(x_i)$ be $1/2$. For $j \in [2 : d]$, if $(x_i, v_1, v_j)$ is labeled $(x_i, v_1^+, v_j^-)$, then we select the the $j$-th coordinate of $g(x_i)$ to be $0$, and otherwise we select it to be $1$. We map $x_i$ to $f(x_i) := g(x_i)/\|g(x_i)\|$.

Next, we prove that all queries are satisfied: the summations for $\|f(x_i) - e_1\|_2^2$ and $\|f(x_i) - e_j\|_2^2$ differ only in the first and the $j$-th coordinates. When the $j$-th coordinate is $0$, we have

$$\|f(x_i) - e_1\|_2^2 - \|f(x_i) - e_j\|_2^2 = \left(1 - \frac{1}{2\|g(x_i)\|_2}\right)^2 - \left(\left(\frac{1}{2\|g(x_i)\|_2}\right)^2 + 1^2\right) = -\frac{1}{\|g(x_i)\|_2} < 0,$$

Hence $(x_i, v_1^+, v_j^-)$ is satisfied. On the other hand, when the $j$-th coordinate is $1$, we have

$$\|f(x_i) - e_1\|_2^2 - \|f(x_i) - e_j\|_2^2 = \left(\left(1 - \frac{1}{2\|g(x_i)\|_2}\right)^2 + 1^2\right) - \left(\frac{1}{2\|g(x_i)\|_2}\right)^2 = 2 - \frac{1}{\|g(x_i)\|_2} > 0,$$

where the inequality holds since $\|g(x_i)\|_2 > \frac{1}{2}$(due to the first coordinate and the $j$-th coordinate). Hence $(x_i, v_j^+, v_1^-)$ is satisfied. Therefore, we can construct $f(x_i)$ that satisfies all the queries with anchor $x_i$. Therefore, we can shatter the set of all $(d-1)(n-d)$ queries.

$\square$

**Theorem D.4** (Class distance). *Consider a domain $V$ partitioned into any number of disjoint classes $C_1, \ldots, C_m$, where $m \geq 2$. For the class indicator function $\rho_C \colon V \times V \to \{0, 1\}$, defined as $\rho_C(x, y) = 0$ iff there exists $i$ such that $x, y \in C_i$, the sample complexity of contrastive learning is $S_3(\epsilon, \delta) = \Theta\left(\frac{n}{\epsilon} \operatorname{polylog}\left(\frac{1}{\epsilon}, \frac{1}{\delta}\right)\right)$ in the realizable case and $\Theta\left(\frac{n}{\epsilon^2} \operatorname{polylog}\left(\frac{1}{\epsilon}, \frac{1}{\delta}\right)\right)$ in the agnostic case.[9]*

*Proof.* We prove this by showing an $\Theta(n)$ bound on the VC dimension of contrastive learning with class distances.

**Upper bound.** Consider any set of samples $\{(x_i, y_i, z_i)\}_{i=1 \ldots k}$ of size $k \geq n$. Fix an arbitrary labeling of this set. Note that in any triple, one pair (e.g. $(x_i, y_i)$) is from the same class and the other (e.g. $(x_i, z_i)$) is from different classes. Create a graph on $n$ vertices, where each vertex corresponds to an element in the dataset. For each labeled sample create an edge in this graph corresponding to the pair of vertices, which are from the same class. Since this graph has $n$ edges, there must be a cycle $\mathcal{C}$ in this graph and all the vertices on this cycle must be in the same class according to the labeling. Consider any edge $(x, y)$ on the cycle and change the labeling of the triple corresponding to this edge. This forces $x$ and $y$ to be from different classes and leads to a contradiction, since $x$ and $y$ must be in the same class due to the existence of a path between them.

**Lower bound.** Partition $V$ into disjoint sets of size 3, and associate a query with each set, i.e.

$$Q = \{(v_1, v_2, v_3), (v_4, v_5, v_6), \ldots, (v_{n-2}, v_{n-1}, v_n)\}.$$

For each labeled query $(v_i, v_{i+1}^+, v_{i+2}^-) \in \vec{Q}$, place $v_i$ and $v_{i+1}$ in $C_1$, and $v_{i+2}$ in $C_2$. For each labeled query $(v_i, v_{i+2}^+, v_{i+1}^-) \in \vec{Q}$, place $v_i$ and $v_{i+2}$ in $C_1$, and $v_{i+1}$ in $C_2$. $\qquad \square$

By Theorem C.3, our bounds extend to the quadruple contrastive learning setting.

**Corollary D.5.** *The same bounds as in Theorem 3.1, Theorem 3.2, Theorem 2.7, Theorem D.2, Theorem D.3 hold for quadruplet contrastive learning.*

# E  LEARNING LABELS UNDER A WELL-SEPARATED ASSUMPTION

In this section, we show that under a well-separated setting (i.e. when the two distances of each triplet are guaranteed to be separated by some multiplicative factor), the sample complexity can be improved to be almost independent of $d$ (disregarding polylog factors).

More formally, given parameter $\alpha > 0$, each labeled query $(x, y^+, z^-)$ implies the constraint $(1 + \alpha)\rho(x, y) < \rho(x, z)$.

**Lemma E.1.** *Let $n, d$ be integers and $0 < \alpha < 1$. Let $T(n, d)$ be the VC dimension of the contrastive learning problem and $T(n, d, \alpha)$ be the VC dimension of the $(1 + \alpha)$-separate contrastive learning problem.*

$$T(n, d, \alpha) = O\big(\min(n^2, nd, n \log n / \alpha^2)\big), \quad T(n, d, \alpha) = \Omega\big(\min(n^2, nd, n/\alpha)\big).$$

We start with the lower bound in Theorem E.1. Recall the embedding $f$ given in the proof of Theorem B.1. From this embedding it follows that $T(n, d, c/d) = \Omega(nd)$ for a sufficiently small constant $c > 0$. Indeed, for all $i \leq n - d$ and $j \leq d$, if $(x_i, v_1^+, v_j^-)$ is satisfied then

$$\|f(x_i) - f(v_j)\|_2^2 - \|f(x_i) - f(v_1)\|_2^2 \geq 1 \geq \|f(x_i) - f(v_1)\|_2^2 / d,$$

where the last inequality follows as all the coordinates are bounded by 1. Rearranging we obtain

$$\|f(x_i) - f(v_j)\|_2 \geq \sqrt{1 + 1/d} \cdot \|f(x_i) - f(v_1)\|_2 \geq (1 + c/d)\|f(x_i) - f(v_1)\|_2.$$

Similarly we have

$$\|f(x_i) - f(v_1)\|_2 \geq (1 + c/d)\|f(x_i) - f(v_j)\|_2.$$

when $(x_i, v_j^+, v_1^-)$ is satisfied. This shows that $T(n, d, c/d) = \Omega(nd)$.

---

[9] For this distance function we also allow an "equality" label $(x, y^+, z^+)$, which implies $\rho_C(x, y) = \rho_C(x, z)$.

Next, when $\alpha \leq c/d$ we have that $T(n, d, \alpha) \geq T(n, d, c/d) = \Omega(nd)$. When $\alpha \geq c/d$ we have that $T(n, d, \alpha) \geq T(n, d_1, c/d_1) = \Omega(n/\alpha)$, where $d_1 := \lceil c/\alpha \rceil$.

Next, we prove the upper bound. We first show a dimension reduction argument, which intuitively shows that in separate contrastive learning can always be reduced to $O(\log n/\alpha^2)$ dimensions.

**Lemma E.2.** *Let $d_1 = \lceil 1000 \log n/\alpha^2 \rceil$. We have that*

$$T(n, d, \alpha) \leq T(n, d_1, \alpha/2).$$

The lemma is a simple application of the Johnson-Lindenstrauss lemma (Johnson, 1984), given below.

**Fact E.3** (The Johnson-Lindenstrauss lemma)**.** *For any $1 > \beta > 0$, a set $X$ of $n$ points in $\mathbb{R}^d$ and an integer $d_1 \geq 15 \log n/\beta^2$, there is an embedding $f : X \to \mathbb{R}^{d_1}$ such that for all $x, y \in X$ we have*

$$(1 - \beta)\|x - y\|_2 \leq \|f(x) - f(y)\|_2 \leq (1 + \beta)\|x - y\|_2.$$

We can now prove Lemma E.2.

*Proof of Lemma E.2.* Suppose that $T(n, d, \alpha) \geq m$ for some integer $m$. By definition, there exists a set of queries $Q = \{(u_i, v_i, w_i) : i \leq m\}$ such that for any labeling of the queries $\vec{Q}$, there is an embedding $f : V \to \mathbb{R}^d$ such that if the query $(u_i, v_i, w_i)$ is labeled as $(u_i, v_i^+, w_i^-)$ then $\|f(u_i) - f(w_i)\|_2 \geq (1 + \alpha)\|f(u_i) - f(v_i)\|_2$. Next, we use the Johnson-Lindenstrauss lemma in order to embed the points $\{f(v) : v \in V\}$ in a lower dimensional space without distorting the distances too much. By Fact E.3 with $\beta := \alpha/7$ we have an embedding $g : V \to \mathbb{R}^{d_1}$ such that if the query $(u_i, v_i, w_i)$ is labeled as $(u_i, v_i^+, w_i^-)$ then

$$\begin{aligned}
\|g(u_i) - g(w_i)\|_2 &\geq (1 - \beta)\|f(u_i) - f(w_i)\|_2 \geq (1 + \alpha)(1 - \beta)\|f(u_i) - f(v_i)\|_2 \\
&\geq \frac{(1 + \alpha)(1 - \beta)}{1 + \beta}\|g(u_i) - g(v_i)\|_2 \geq (1 + \alpha/2)\|g(u_i) - g(v_i)\|_2,
\end{aligned} \tag{1}$$

where the last inequality holds for all $0 < \alpha < 1$. This shows that $T(n, d_1, \alpha/2) \geq m$ and completes the proof. $\square$

Given this lemma, our claim follows since

$$T(n, d, \alpha) \leq T(n, d_1, \alpha/2) \leq T(n, d_1) = O(nd_1) = O(n \log n/\alpha^2).$$

Where the first inequality holds due to Lemma E.2, the second since for any integers $n', d'$ and value $\alpha' > 0$ we have that $T(n', d', \alpha') \leq T(n', d')$ (i.e. the exact version is at least as hard to learn as the separate version, since the hypothesis space of the latter is contained in the former), and the third equality holds due to Theorem 3.2. This concludes the proof of Lemma E.1.

Using Lemma 2.6, we obtain our sample complexity bounds.

**Theorem E.4.** *For any $1 > \alpha > 0$, the sample complexity of $(1 + \alpha)$-separate contrastive learning is $S_3(\epsilon, \delta) = O\left(\frac{n \log n}{\alpha^2 \cdot \epsilon} \text{polylog}(\frac{1}{\epsilon}, \frac{1}{\delta})\right)$ and $S_3(\epsilon, \delta) = \Omega\left(\frac{n}{\alpha \cdot \epsilon} \text{polylog}\left(\frac{1}{\epsilon}, \frac{1}{\delta}\right)\right)$ for the realizable case, and $S_3(\epsilon, \delta) = O\left(\frac{n \log n}{\alpha^2 \cdot \epsilon^2} \text{polylog}\left(\frac{1}{\epsilon}, \frac{1}{\delta}\right)\right)$ and $S_3(\epsilon, \delta) = \Omega\left(\frac{n}{\alpha \cdot \epsilon^2} \text{polylog}\left(\frac{1}{\epsilon}, \frac{1}{\delta}\right)\right)$ for the agnostic case.*

We note that the case of $\alpha \geq 1$ is a relaxed version of e.g. $\alpha = 0.99$, therefore we immediately obtain the following near-tight bounds:

**Corollary E.5.** *For any $\alpha \geq 1$, the sample complexity of $(1 + \alpha)$-separate contrastive learning is $S_3(\epsilon, \delta) = O\left(\frac{n \log n}{\epsilon} \text{polylog}(\frac{1}{\epsilon}, \frac{1}{\delta})\right)$ and $S_3(\epsilon, \delta) = \Omega\left(\frac{n}{\epsilon} \text{polylog}\left(\frac{1}{\epsilon}, \frac{1}{\delta}\right)\right)$ for the realizable case, and $S_3(\epsilon, \delta) = O\left(\frac{n \log n}{\epsilon^2} \text{polylog}\left(\frac{1}{\epsilon}, \frac{1}{\delta}\right)\right)$ and $S_3(\epsilon, \delta) = \Omega\left(\frac{n}{\epsilon^2} \text{polylog}\left(\frac{1}{\epsilon}, \frac{1}{\delta}\right)\right)$ for the agnostic case.*

As an interesting open problem, we leave open whether the upper bound of $O(n \log n)$ can be improved for large enough values of $\alpha$.

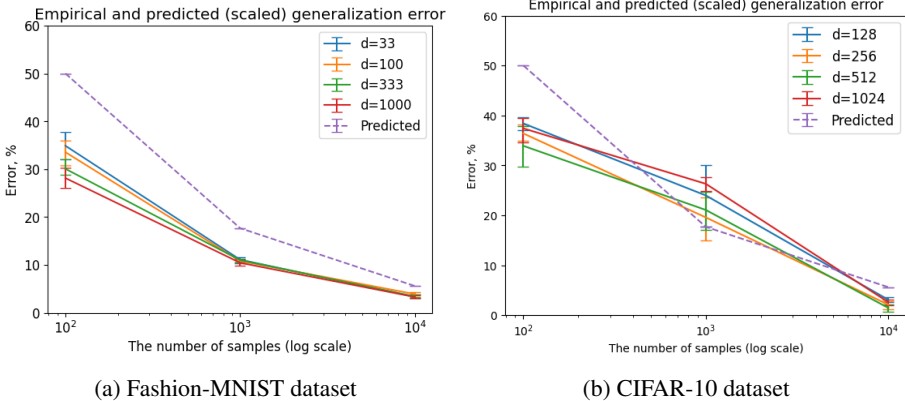

(a) Fashion-MNIST dataset        (b) CIFAR-10 dataset

Figure 1: Training with triplet loss. For various embedding dimensions $d$, we show the empirical and the scaled predicted generalization errors. The scaling factor is chosen as $\sqrt{1/320}$ (Mohri et al., 2018). In this scenario, we consider the well-separated case, and hence the predicted error is the same for all $d$. Data points are averaged over 10 runs, error bars show $10\%$ and $90\%$ quantiles.

## F  EXPERIMENTS

In this section, we empirically verify our theoretical findings on real-world image datasets. We compute image embeddings using the ResNet-18 network, with the last layer being replaced with a linear layer with the output dimension matching that of the target embedding dimension. The neural network is trained from scratch for 100 epochs using a set of $m \in \{10^2, 10^3, 10^4\}$ training samples, and is evaluated on a different test set of $10^4$ triplets from the same distribution. We show that our theory provides a good estimation of the gap between the training and test error.

We consider the standard experimental setup used in the literature where the positive example is sampled from the same class as the anchor, and negative examples are sampled from a different class, see e.g. Saunshi et al. (2019); Awasthi et al. (2022). Note that it means that the ground-truth distances are well-separated, and hence we are in the setting of Theorem 4.2 (see Section F for experiments in the non-well-separated case). Since the neural network doesn't perfectly fit the data, the setting is agnostic, and hence for $n$ data points and error parameter $\epsilon$, our theory predicts that $m \approx c\frac{n}{\epsilon^2}$ labeled samples are required, where we estimate $c = 320$ based on Mohri et al. (2018, Page 48). Hence, given $m$ samples, our theory predicts that $\epsilon \approx \sqrt{\frac{n}{320m}}$. We compare this value to the empirical generalization error $\tilde{\epsilon}$, defined as the difference between training and test errors.

We consider two settings: when each sample has a single negative example and when it has multiple negative examples.

**Single negative example:** When every sample has only one negative, we train the model from scratch on CIFAR-10 (Krizhevsky, 2009) and Fashion-MNIST (Xiao et al., 2017) datasets using the marginal triplet loss (Schroff et al., 2015b)

$$L_{MT}(x, y^+, z^-) = \max(0, \|x - y^+\|^2 - \|x - z^-\|^2 + 1),$$

where $x$ and $y^+$ are points from the same class and $z^-$ is a point from a different class.

We present our results in Figure 1. For a small number of samples, there is a discrepancy between the predicted and the empirical generalization errors, which is due to the fact that the random guess trivially achieves $50\%$ accuracy, while our results hold only for the non-trivial error rates (see Definition 2.1). First, the results for different embedding dimensions match closely, which shows that the error doesn't depend on the dimension, as predicted by our theory. Note that the scaled predicted and the empirical results are within a constant factor, which is hard to attribute to coincidence given the choice of the scaling factor. Hence, these experiments support the theory by showing that it provides a good predictor of practical generalization error.

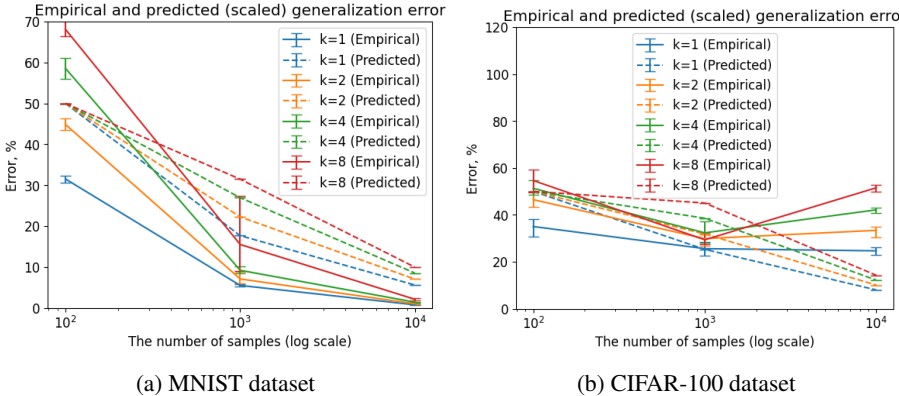

(a) MNIST dataset                    (b) CIFAR-100 dataset

Figure 2: Training with the contrastive loss with $k \in \{1, 2, 4, 8\}$ negatives. We show the empirical and the scaled predicted generalization errors. The data points correspond to the average over 10 runs, and the error bars show $10\%$ and $90\%$ quantiles

**Multiple negative examples:** For the case of multiple negative samples, we train the model using the contrastive loss (Logeswaran & Lee, 2018b)

$$L_C(x, y^+, z_1^-, \ldots, z_k^-) = -\log \frac{\exp(x^T y^+)}{\exp(x^T y^+) + \sum_{i=1}^{k} \exp(x^T z_i^-)}.$$

We train the model from scratch on the MNIST (Yann, 1998) and CIFAR-100 (Krizhevsky, 2009) datasets, and report $\tilde{\epsilon}/\epsilon$ for different numbers of negatives $k \in [1, 2, 4, 8]$. Recall that for $k$ negatives we have $\epsilon \in \Omega(\frac{n}{\epsilon^2}) \cap O(\frac{n}{\epsilon^2} \log(k+1))$, and for our experiments we use the upper bound. The results are shown in Figure 2. As before, the values are within a constant factor for all values of $k$ and numbers of samples $m$. Figure 2(b) shows that the $\log(k+1)$ factor in the sample complexity is observable in practice (we use it when computing the predicted error and the curves for different $k$ match closely after this normalization). The lack of convergence in Figure 2(b) is most likely attributable to the fact that the sample size is too small. Similarly to other plots, we expect convergence for larger sample sizes, which were computationally prohibitive to include in this version.

**Non-well-separated case** Finally, we conduct experiments for the case when the positive example is not necessarily sampled from the same class as the anchor. While the experiments above correspond to one of the most conventional contrastive learning approaches, the experiments in this section more closely match our theoretical settings. In particular, we don't use any data augmentation for the training, and we use the same set of points (but not the same set of triplets) for both training and testing. Since the VC-dimension depends on the number of points (Theorems 3.1 and 3.2), to verify those results, we subsample $n \in \{10^2, 10^3, 10^4\}$ points.

In contrast with the previous experiments, where the positive example is sampled from the same class as the anchor, in this section, all elements of a triplet are sampled uniformly from the chosen $n$ points. We determine negative and positive examples based on the distance between ground-truth embeddings computed using a pretrained ResNet-18 network.

We compare the empirical generalization error $\tilde{\epsilon}$ to the theoretically predicted error rate $\epsilon$ (compared to the best classifier). We perform experiments on the training set of CIFAR-10 and the validation set of ImageNet by training ResNet-18 from scratch on $m \in \{2, 10, 10^2, 10^3, 10^4, 10^5\}$ randomly sampled triplets, and evaluating the model on the $10^4$ triplets sampled from the same distribution[10]. In Figure 3, the predicted and the empirical errors show the same tendencies and the same relative behavior between different choices of $n$. As before, the empirical and the scaled predicted errors are within a factor of 2 of each other.

---

[10]We express our thanks to the FFCV library (Leclerc et al., 2022) which allowed us to significantly speed up the execution

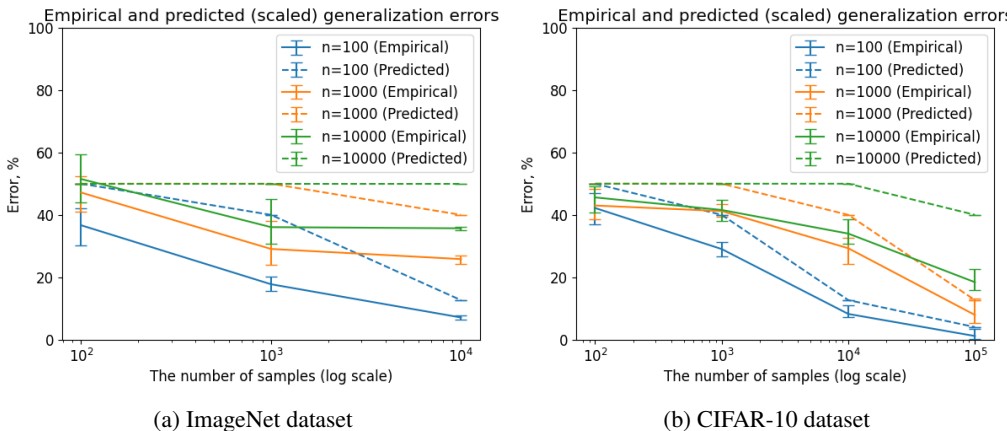

(a) ImageNet dataset

(b) CIFAR-10 dataset

Figure 3: Training with the contrastive loss with one negative with embedding dimension $512$. For each $n \in \{10^2, 10^3, 10^4\}$, we subsample $n$ points from a dataset for $m \in \{10^2, 10^3, 10^4, 10^5\}$ input triplets. For various $n$, we show the empirical and the scaled predicted generalization errors. Note that the theoretical predicted error is bounded by $50\%$, which is a achieved by a random guess. Data points are averaged over 10 runs, error bars show $10\%$ and $90\%$ quantiles.

