# OpenReview forum: "Optimal Sample Complexity of Contrastive Learning"
_ICLR.cc/2024/Conference — ICLR 2024 spotlight_

### Official Review · Reviewer_YPGd · 2023-10-29

**Soundness:** 4 excellent
**Presentation:** 4 excellent
**Contribution:** 3 good
**Rating:** 8
**Confidence:** 3

**Summary:**

Suppose $V$ is a set of $n$ data points, each embedded into a $mathbb{R}^d$. Suppose we observe $m$ triplets $(x,y,z) \in V^3$ and their labels in $\{ -1, 1\}$ where the label is $1$ if $x,y$ are closer than $x,z$ in the embedding space in $\ell_p$ distance and $-1$ otherwise. The paper gives optimal order for sample complexity $m$ required to get to a misclassification error of $\epsilon$.

The high level technique is to derive VC dimension (and Natarajan dimension for larger tuples, but we ignore results in the larger tuple case for now). The authors derive nearly tight order for VC dimension of such triplet classifiers. For upper bounding the VC dimension, they formulate the classification function as the sign of a polynomial in $nd$ dimensions. The key ingredient here is a fact from Warren (1968) that there are at most $(4epm/nd)^{nd}$ connected components in $\mathbb{R}^{nd}$ where in each connected component, the signs of the $m$ polynomials are fixed. For proving the lower bound on VC dimension, the authors give a clever construction of a set of triplets which can be shattered.

Authors give their results to realizable and agnostic cases. They extend their results to several distance functions and tuples of size more than 3. They also study the well-separated case where the labeled triplets $(x,y^+,z^-)$ satisfy $\rho(x,z) \geq (1+\alpha) \rho(x,y)$ etc. for some $\alpha > 0$.

**Strengths:**

* The derivations are interesting, short and non-trivial.
* The results are relevant because contrastive learning is practical.

**Weaknesses:**

Nothing significant.

**Questions:**

Typos/Minor comments:
* page 4: Outline of techniques: P is a polynomial of degree $2$, not $2d$.
* Reducing dependence on $n$: It may be good to state bounds with the assumptions mentioned ($k$ latent classes etc.)
* Theorem 3.3 proof, first line: Should it be $d < n$ here?
* In Definition 2.1, the symbol $S_3$ seems to be used before definition.
* Please search for "Kulis Kulis" and "Warren Warren" in the paper and remove such duplicates.
* page 8, second line: Should $<$ be replaced by $\leq$?

---

> ### Author Response · Authors · 2023-11-19
>
> We thank the reviewer for their comments. We updated the PDF, which now includes all of the suggested changes.
>
> **Q**: Typos:
> * page 4: Outline of techniques: P is a polynomial of degree $2$, not $2d$.
> * Theorem 3.3 proof, first line: Should it be $d<n$ here?
> * Please search for "Kulis Kulis" and "Warren Warren" in the paper and remove such duplicates.
> * page 8, second line: Should $<$ be replaced by $\le$?
>
> **A**: Thank you, fixed!
>
> **Q**: Reducing dependence on n: It may be good to state bounds with the assumptions mentioned (k latent classes etc.)
>
> **A**: In case there are $k$ latent classes, all the results would be as in Table 1, with $n$ being replaced with $k$.
>
> **Q**: In Definition 2.1, the symbol $S_3$ seems to be used before definition.
>
> **A**: Thank you for pointing it out. Definition 2.1 introduces the notion of sample complexity, and also introduces notation $S_3$ for the sample complexity. We clarified the wording: “the sample complexity of contrastive learning, denoted as $S_3(\epsilon, \delta)$”

---

### Official Review · Reviewer_Tspf · 2023-11-01

**Soundness:** 2 fair
**Presentation:** 2 fair
**Contribution:** 3 good
**Rating:** 6
**Confidence:** 3

**Summary:**

The paper explores the efficacy of contrastive learning, a method for learning data representations based on labeled tuples that detail distance relationships within the tuples. The main focus is on understanding the sample complexity of this method, which refers to the minimum number of labeled tuples needed to achieve accurate generalization.

This work provides specific bounds for sample complexity across various settings, especially for arbitrary distance functions, $\ell_p$-distances, and tree metrics. A central finding is that for learning $\ell_p$-distances, a minimum of $\Theta(\min(nd,n^2))$ labeled tuples is sufficient and necessary for depicting $d$-dimensional representations of $n$-point datasets. These results are applicable regardless of the input samples' distribution and derive from bounds on the Vapnik-Chervonenkis/Natarajan dimension of related problems.

This paper also demonstrates that theoretical boundaries derived from the VC/Natarajan dimension correlate strongly with experimental outcomes.

**Strengths:**

1. This paper primarily studies the sample complexity of contrastive learning, and provides tight bounds in some settings, including arbitrary distance functions and $\ell_p$-distances for even $p$ and almost tight bounds for odd $p$. For constant $d$, this work also provides a matching upper and lower bound.
2. This paper studies both realizable and agnostic settings, as is standardly considered in PAC learning. The sample complexity bounds in terms of $\epsilon$ coincide with the standard results in PAC learning in both realizable and agnostic settings.
3. The proposed proof idea extends to various settings, including the cases where $k >1$ (multiple negative examples in one tuple) and quadruple samples.

**Weaknesses:**

1. It would be good to provide a thorough comparison with the known sample complexity bounds that appeared in the existing literature.
2. While I understand the page limit of the main body, there seems to be relatively less than enough content on the main results of this work in the main body. Perhaps consider moving more technical parts from the appendix into the main body.
3. The structure of the paper could be reorganized a bit: e.g., the paragraph "Reducing dependence on $n$" could be part of the discussions after presenting the full main results.

**Questions:**

1. What do you think is the primary season/insight that the upper bounds for $\ell_p$-distances are different between odd and even $p$?
2. Do you think it is possible to characterize the sample complexity bound when the cardinality of $V$ is infinite?
3. Minor:
- In the paragraph "Outline of the techniques" on Page 4, why "P is some polynomial of degree $2d$"?
- In the paragraph above "Outline of the techniques" on Page 4, $(x_1^-, x_2^-)$ ->  $(x_3^-, x_4^-)$?

---

> ### Author Response · Authors · 2023-11-19
>
> We thank the reviewer for their comments. We updated the PDF, which now includes all of the suggested changes. We would to highlight that:
> * as you correctly mentioned in the summary, our work provides distribution-independent assumptions, which separates our work from other related works that make various assumptions about the data, model, or training process;
> * our bounds are achieved using PAC-learning techniques, which, contrary to the widespread belief, demonstrates that PAC-learning techniques are applicable to the contrastive learning setting.
>
> We also would be very grateful if you could clarify why soundness was estimated as “2. Fair”. Please let us know if you have any specific questions about the proof of our results.
>
> **Q**: A thorough comparison with the known sample complexity bounds.
>
> **A**: We first would like to emphasize that our setting is substantially different from all the previous work we are aware of: namely, we estimate the sample complexity without any assumptions on data distribution, classifier, or training algorithm. Moreover, multiple works measure the classifier quality using some continuous loss function instead of prediction accuracy (and hence $\epsilon$ is formulated in terms of the loss function), which again makes the results incomparable. We below outline the works on sample complexity which are most related to ours:
> * “Sample complexity of learning Mahalanobis distance metrics” by N. Verma and K. Branson shows that learning a $n \times n$ Mahalanobis matrix requires $O(\frac{n}{\epsilon^2})$ samples.
> * “Tree Learning: Optimal Sample Complexity and Algorithms” by Avdiukhin et al. (AAAI 2023) is an example of the complexity bound for the specific metric case, showing that the VC dimension of tree learning is $\Theta(n)$. The goal of tree learning is to build a hierarchy on $n$ points such that constraints of the form “$c$ is separated from $a$ and $b$ first” are satisfied. This is a metric case where the distance between $a$ and $b$ can be defined as the number of leaves under the least common ancestor of $a$ and $b$.
> * Saunshi et al. (2019) provide sample complexity bounds for transfer learning settings.
> * A line of work considers the sample complexity of recovering metric embeddings using noisy queries in various settings. In particular, “Landmark Ordinal Embedding” by Ghosh et al. shows that $O(d^8 n \log n)$ noisy triplet queries suffice to achieve a constant additive divergence.
>
> **Q**: While I understand the page limit of the main body, there seems to be relatively less than enough content on the main results of this work in the main body. Perhaps consider moving more technical parts from the appendix into the main body.
>
> **A**: Thank you for the suggestion, we moved the proof of Theorem B.2 (constant $d$ case) to the main body. Given the page limit, while other technical results can’t be included in the main body, we give an outline of the proofs in the “Outline of the techniques”.
>
> **Q**: The structure of the paper could be reorganized a bit: e.g., the paragraph "Reducing dependence on n" could be part of the discussions after presenting the full main results.
>
> **A**: Thank you, we moved it to the “Extensions” section.
>
> **Q**: What do you think is the primary reason/insight that the upper bounds for $\ell_p$-distances are different between odd and even p?
>
> **A**: The main reason that the upper bounds for odd $p$ are different from the upper bounds for even $p$ is that, for even $p$, $\|x - y\|_p^p$ is a polynomial in $x_1,\ldots,x_d,y_1, \ldots, y_d$, while for odd $p$ it is not, and it is not even a $p$-times continuously differentiable. Intuitively, $\|x - y\|_p^p$ is a worse-behaving function for odd $p$ than for even $p$, so it is not surprising that the upper bound is similarly worse.
>
> **Q**: Is it possible to characterize the sample complexity bound when the cardinality of $V$ is infinite?
>
> **A**: We briefly outline the idea in the paragraph “Reducing dependence on $n$”. The following are natural scenarios.
> * In practice, in order to reduce $n$ one can use any unsupervised method which allows one to perform deduplication without substantially affecting the metric structure. For example, an extremely common assumption in the literature is that the data comes from $k \ll n$ different classes. In this case, one can consider using an unsupervised clustering algorithm (e.g. a pretrained neural network) to partition the points into $k$ clusters, effectively replacing $n$ with $k$ in all bounds.
> * Another practical scenario is that the distribution is supported on a large domain but contains a large number of low-probability outliers. In this case, one can replace $n$ with the support of 99% of the distribution at the cost of a small increase in error.
>
> **Q**: "Outline of the techniques": why "P is some polynomial of degree 2d"?
>
> **A**: Thank you, this is a typo - P is a polynomial of degree 2.
>
> **Q**: $(x_1^−,x_2^−)  \to (x_3^−,x_4^−)$
>
> **A**: Thank you, fixed!

---

> > ### Author Response · Authors · 2023-11-21
> >
> > Dear reviewer,
> >
> > As the discussion period is concluding tomorrow, we would like to ask if our reply has addressed all of your questions about our work.

---

> > ### Comment · Reviewer_Tspf · 2023-11-22
> >
> > Thanks for the responses. Yes, I think my questions about this paper are addressed, and I keep the positive score for this work.

---

### Official Review · Reviewer_m7mJ · 2023-11-01

**Soundness:** 4 excellent
**Presentation:** 2 fair
**Contribution:** 3 good
**Rating:** 8
**Confidence:** 3

**Summary:**

Given labeled sample $(x_1,y^+_1,z^-_1),\ldots,(x_n,y^+_n,z^-_n)$, the goal of contrastive learning is to create a distance function $\rho$ such that $\rho(x,y) < \rho(x,z)$. This study comes in theoretical flavour, providing lower and upper bound for sample complexity of contrastive learning via PAC-learning framework. The main ingredient of the proof of the bound is the Natarajan dimension (which is a generalization of the VC dimension) and the results from Ben David et al. (1995).

Reference:
S. Bendavid, N. Cesabianchi, D. Haussler, P.M. Long, Characterizations of Learnability for Classes of {0, ..., n)-Valued Functions, Journal of Computer and System Sciences, Volume 50, Issue 1, 1995, Pages 74-86.

**Strengths:**

- The study covers a wide range of distance functions, both in lower bounds and upper bounds.
- Interesting use of an algebraic geometry result to prove the upper bound.
- The theory is well-supported by the empirical results.
- The authors discuss possible directions for future work.

**Weaknesses:**

Personally, I find it hard to follow the results by chapter. For example, Section 3 should be all about the bounds in $\ell_p$ norm, and so Theorem 3.1 (Arbitrary distance) should come before this section. And I think it would be easier to follow if Section 3 is only for $k=1$ and Section 4 is only for $k>1$.

I have some comments for Table 1:
- I would put "$(1+\alpha)$-seperable $\ell_2$" in the same category as $\ell_p$.
- The "Same" labels for quadruplet learning and $k$ negatives have different meaning; while the former refer to all distance functions above, the latter only refers to the $\ell_p$ distances. Could the authors modify the table so that this distinction becomes clearer?

**Questions:**

See Weaknesses.

---

> ### Author Response · Authors · 2023-11-19
>
> We thank the reviewer for their comments. We updated the PDF, which now includes all of the suggested changes. In addition to the strengths of our work you identified, we would like to highlight the following important aspects of our work:
> * our work shows that, contrary to the widespread belief, PAC-learning can provide useful bounds for sample complexity in deep learning, specifically in the contrastive learning setting;
> * our work provides bounds that are independent of any assumptions on the input or the learning algorithm.
>
> **Q**: Personally, I find it hard to follow the results by chapter. For example, Section 3 should be all about the bounds in $\ell_p$ norm, and so Theorem 3.1 (Arbitrary distance) should come before this section. And I think it would be easier to follow if Section 3 is only for $k=1$ and Section 4 is only for $k>1$.
>
> **A**: Thank you, we moved Theorem 3.1 to the previous section. We moved the results from subsection 3.1 to a separate section.
>
> **Q**: I would put "(1+$\alpha$)-separable $\ell_2$" in the same category as $\ell_p$.
>
> **A**: Fixed
>
> **Q**: The "Same" labels for quadruplet learning and k negatives have different meaning; while the former refer to all distance functions above, the latter only refers to the ℓp distances. Could the authors modify the table so that this distinction becomes clearer?
>
> **A**: Thank you for pointing this out. We’ve clarified in Table 1 that our results for k negatives in fact cover all distances and not just ℓp (the results are in Appendix C.1).

---

### Official Review · Reviewer_MJ3x · 2023-11-01

**Soundness:** 4 excellent
**Presentation:** 4 excellent
**Contribution:** 3 good
**Rating:** 8
**Confidence:** 2

**Summary:**

This paper studies the sample complexity of contrastive learning, which learns the similarity (usually the distance in a metric space) between domain points, given tuples each labeling a most-similar input point to a given point (anchor).

This paper proves matching (or some almost matching) bounds on the sample complexity for contrastive learning of different metrics (arbitrary distance, cosine similarity, and tree metric), with generalization to learning with hard negatives (separated $\ell_2$ distance), quadruplet learning, or learning with $k$ negatives.

The results are based on an output-based assumption: there is an embedding into a $d$-dimensional vector space. This enables reasoning on the VC/Natarajan dimension to study the PAC learning framework, both for the realizable and the agnostic cases, to get non-vacuous PAC-learning bounds with predictive powers.

The proof is on representing the decision boundary of contrastive learning under such metrics as a low-degree polynomial, and upper bounding its number of possible satisfiable sign changes (Lemma 3.5 proved in Section B), hence the largest shattered set of tuples, and VC/Natarajan dimension.

The theoretical result is also experimentally verified on popular image datasets (CIFAR-10/100 and MNIST/Fashion-MNIST), by learning the representations with a ResNet18 trained from scratch with different contrastive losses.

**Strengths:**

The proof is by _understanding the problem via transforming it_ to another equivalent representation: the decision boundary of contrastive learning under common metrics as a low-degree polynomial, and bounding its number of possible satisfiable sign changes. There is no loss until invoking the algebraic-geometric bounds on number of connected components by Warren, and the sample complexity bounds on Natarajan dimension by Ben David et al.

And at a high level, the paper shows that when the learned representation is expressive enough (such as ResNet18), PAC-learning bounds (e.g., by VC/Natarajan dimension) can have predictive powers.

The theoretical result is also experimentally verified on popular image datasets (CIFAR-10/100 and MNIST/Fashion-MNIST), by learning the representations with a ResNet18 trained from scratch with different contrastive losses (Appendix F).

**Weaknesses:**

The proof arguments are somewhat non-constructive, due to using a counting argument/pigeonhole principle, and hence while the theory may explain certain observations (e.g., the experimental results in Appendix F), it is unlikely to give effective learning algorithms and the constants in the resulting bounds are unlikely to be sharpened. This may be nitpicking, but are weaknesses nonetheless.

**Questions:**

While the experimental results (in Appendix F) verify the growth of error rates as predicted by the theory for ResNet18 on certain parameter ranges, there is not much explanations regarding, e.g., what representations (e.g., deep learning architectures) are expressive enough to achieve the sample bounds as predicted by the theory. That is, the representations (given by the theory) are somehow non-explicit/ineffective, is that the case?

### Typos?

Statement of Theorem 1.3 (Page 4):
The sample complexity of contrastive learning ~for contrastive learning~ under cosine similarity is...

---

> ### Author Response · Authors · 2023-11-19
>
> First, we would like to thank the reviewer for their helpful comments. We would like to add that one of the main points of our work lies in the fact that we provide sample complexity bounds without any assumptions on the data distribution or the learning algorithm. We also would like to emphasize that the focus of our work is sample complexity - namely, how many labeled examples are necessary to produce a good classifier, *regardless of the model and the training algorithm*. This question is important by itself since it allows one to determine whether the amount of data available is sufficient to train a good classifier.
>
> **Q**: The proof arguments are somewhat non-constructive, due to using a counting argument/pigeonhole principle, and hence while the theory may explain certain observations (e.g., the experimental results in Appendix F), it is unlikely to give effective learning algorithms and the constants in the resulting bounds are unlikely to be sharpened. This may be nitpicking, but are weaknesses nonetheless.
>
> **A**: The learning algorithm suggested by our work is Empirical Risk Minimization (ERM), i.e. minimizing the error on the sample data (which is optimal for statistical learning problems in general). While theoretically the ERM problem is computationally hard, any algorithm which performs well in practice can be used. Empirical success of deep learning architectures makes them a suitable choice for the problems we consider. The focus of our work is to quantify the amount of data required for fitting such architectures to the training data while providing generalization guarantees.
>
> **Q**: While the experimental results (in Appendix F) verify the growth of error rates as predicted by the theory for ResNet18 on certain parameter ranges, there is not much explanations regarding, e.g., what representations (e.g., deep learning architectures) are expressive enough to achieve the sample bounds as predicted by the theory. That is, the representations (given by the theory) are somehow non-explicit/ineffective, is that the case?
>
> **A**: One property of the neural network architecture that we can analyze is the dimension of the embedding layer, denoted as $d$. In particular:
> Using our results, given the number of available samples, one can estimate whether they are sufficient for achieving low generalization error. In particular, our work shows that for networks with a Euclidean embedding layer of dimension $d$, approximately $nd$ samples are necessary and sufficient where n is the number of different points in the domain.
> Furthermore, in the well-separated case we show that the number of samples required is independent of the dimension of the embedding layer.

---

### Meta-Review · Area_Chair_dETn · 2023-12-08

**Metareview:**

This paper was uniformly appreciated by all reviewers, and provides novel insights into the important problem of contrastive learning. Clear accept.

**Justification For Why Not Higher Score:**

This could be an oral.

**Justification For Why Not Lower Score:**

All reviewers clearly supported the paper.

---

### Decision · Program_Chairs · 2024-01-16

Accept (spotlight)